# EGFL9 promotes breast cancer metastasis by inducing cMET activation and metabolic reprogramming

Fanyan Meng[1,2,10], Ling Wu [1,10], Lun Dong[1,3,10], Allison V. Mitchell[1], C. James Block[1], Jenney Liu[4], Haijun Zhang[5], Qing Lu[6], Won-min Song[7], Bin Zhang[7], Wei Chen[1], Jiani Hu[8], Jian Wang[9], Qifeng Yang [3], Maik Hüttemann[4] & Guojun Wu [1]*

The molecular mechanisms driving metastatic progression in triple-negative breast cancer (TNBC) patients are poorly understood. In this study, we demonstrate that epidermal growth factor-like 9 (EGFL9) is significantly upregulated in basal-like breast cancer cells and associated with metastatic progression in breast tumor samples. Functionally, EGFL9 is both necessary and sufficient to enhance cancer cell migration and invasion, as well as distant metastasis. Mechanistically, we demonstrate that EGFL9 binds cMET, activating cMET-mediated downstream signaling. EGFL9 and cMET co-localize at both the cell membrane and within the mitochondria. We further identify an interaction between EGFL9 and the cytochrome *c* oxidase (COX) assembly factor COA3. Consequently, EGFL9 regulates COX activity and modulates cell metabolism, promoting a Warburg-like metabolic phenotype. Finally, we show that combined pharmacological inhibition of cMET and glycolysis reverses EGFL9-driven stemness. Our results identify EGFL9 as a therapeutic target for combating metastatic progression in TNBC.

[1] Barbara Ann Karmanos Cancer Institute, Department of Oncology, Wayne State University School of Medicine, 4100 John R, Detroit, MI 48201, USA. [2] Comprehensive Cancer Centre of Drum Tower Hospital, Medical School of Nanjing University and Clinical Cancer Institute of Nanjing University, 321 Zhongshan Road, 210008 Nanjing, China. [3] Department of Breast Surgery, Qilu Hospital, Shandong University, Jinan, Shandong Province, China. [4] Center for Molecular Medicine and Genetics, Wayne State University, Detroit, MI 48201, USA. [5] College of Life Sciences, Huaibei Normal University, 235000 Huaibei, Anhui Province, China. [6] Department of Radiology, Renji Hospital, School of Medicine, Shanghai Jiaotong University, 160 Pujian Rd, 200127 Shanghai, China. [7] Department of Genetics and Genomic Sciences, Icahn Institute of Genomics and Multiscale Biology, Icahn Mount Sinai School of Medicine, New York, NY 10029, USA. [8] Department of Radiology, Wayne State University MRI Core, 4201 St. Antoine, Detroit, MI 48201, USA. [9] Department of Pathology, Wayne State University, 550 E. Canfield Road, Detroit, MI 48201, USA. [10] These authors contributed equally: Fanyan Meng, Ling Wu, Lun Dong. *email: wugu@karmanos.org

Metastasis is the main cause of death in breast cancer patients and represents an area of unmet clinical need[1]. Accumulating evidence indicates metastasis is a multistep and multifaceted dynamic process[2]. In addition to genetic mutations, altered gene expression, and aberrant activation of critical signaling pathways, cancer cells exhibit apparent mitochondrial dysfunction during cancer progression and metastasis. Mitochondrial dysfunction is characterized by a loss of efficiency in the electron transport chain and reductions in the synthesis of high-energy molecules, such as adenosine-5′-triphosphate (ATP). These changes lead to defects in oxidative phosphorylation (OXPHOS) and increased glycolysis[3,4]. There is increasing evidence that mitochondrial dysfunction and metabolic reprogramming directly impact gene expression, reactive oxygen species (ROS) production, and calcium homeostasis[4,5]. As a result, changes in metabolic phenotype are closely associated with modulation of cell cycle and cell differentiation, which promotes cancer progression to a metastatic phenotype[6,7]. However, there have been few studies linking pro-metastatic genes or cell signaling pathways to impaired mitochondrial function and metabolic reprogramming.

Members of epidermal growth factor (EGF) repeat superfamily are characterized by the presence of EGF-like repeats and are often involved in the regulation of the cell cycle, proliferation, and developmental processes. EGF and Epiregulin (a member of the EGF family) can function as a ligand of epidermal growth factor receptor (EGFR) and activate EGFR signaling, which will promote multiple biological consequences associated with cancer development. Notch family members are also EGF repeat containing proteins that function as receptors for membrane-bound ligands and may play multiple roles during development and cancer progression, as well as, in regulation of stem cells and cancer stem cells. EGFL6 promotes endothelial cell migration and angiogenesis through the activation of extracellular signal-regulated kinase (ERK)[8]. EGFL6 can increase ovarian tumor growth and metastasis through maintaining ALDH+ stem cell[9]. EGFL7 is expressed in the bone microenvironment and promotes angiogenesis via ERK, STAT3, and integrin signaling cascades[10]. Moreover, EGFL7 enhances EGFR–AKT signaling, epithelial to mesenchymal transition (EMT), and metastasis of gastric cancer cells[11]. EGFL7 was also shown to modulate Notch signaling and affect neural stem cell renewal[12].

In this study, we demonstrate that EGFL9 is the only member of the EGF-like protein family that is highly expressed in basal-like breast cancer cells and significantly correlates with metastatic potential in a set of human breast cancer specimens. EGFL9 significantly promotes cell migration and invasion in vitro, and cancer metastasis in vivo. Mechanistically, we find a physical interaction between EGFL9 and cMET, which activates cMET and downstream signaling pathways related to cancer metastasis. Separately, we demonstrate that EGFL9 translocates to the mitochondria and interacts with COX assembly factor COA3. EGFL9 overexpression promotes significant decreases in COX activity, oxygen consumption rate, and an increase in lactate production and glucose uptake, which suggests that EGFL9 drives a metabolic switch from oxidative phosphorylation to glycolysis. Our results suggest that high expression of EGFL9 and its induction of both cMET signaling activation and metabolic reprogramming is a novel mechanism promoting breast cancer metastasis. We propose EGFL9 signaling as a therapeutic target for developing an effective treatment for metastatic breast cancer.

## Results

**EGFL9 is overexpressed in human breast cancer.** To investigate the expression pattern of EGF repeat superfamily in breast cancer, we determined the expression of nine members of this superfamily in a set of human breast cancer cell lines using real-time RT-PCR. Our results showed that only *EGFL9* was preferentially expressed in basal-like breast cancer cells. In contrast, *EGFL2* showed preferential expression in luminal breast cancer cell lines, while the other members did not show a recognizable pattern (Fig. 1a, Supplementary Fig. 1, Supplementary Table 1). We confirmed the EGFL9 expression pattern in a panel of human breast cancer cell lines. *EGFL9* was highly expressed in most metastatic basal-like cells, while we observed lower expression of *EGFL9* in non- or low-metastatic luminal cell lines (Fig. 1b, c, Supplementary Table 1). Data mining in Oncomine confirmed that *EGFL9* expression was significantly higher in TNBC cell lines than in non-TNBC cell lines (Supplementary Fig. 2a)[13]. In addition, *EGFL9* expression was also significantly higher in basal-like or triple-negative breast tumor samples than non-basal-like or non-TNBC tumor samples (Supplementary Fig. 2b–d)[14–16].

Next, we investigated the expression pattern of EGFL9 in clinical breast tumor samples. We found high expression of EGFL9 in 7/25 (28%) of primary breast tumors from patients with coincident metastasis. In contrast, low expression of EGFL9 was found 23/45 (51.1%) of breast tumors from patients without metastatic disease (Fig. 1d, e). The Cochran-Armitage trend test indicated that the probability of metastasis significantly increased with increased intensity of EGFL9 ($P = 0.02$) (Fig. 1d, e; Supplementary Table 2). Additionally, the probability of a tumor being poorly differentiated (Grade = 3) increased as intensity of EGFL9 increased ($P = 0.06$). No other statistically significant association was identified between EGFL9 expression and other pathological parameters (Supplementary Table 2). Overall, these results suggest that high expression of EGFL9 is related to the metastatic phenotype and basal-like characteristics in breast cancer.

**EGFL9 influences cell migration and invasion in vitro.** To investigate the function of *EGFL9* in cancer metastasis, we established two *EGFL9* overexpression cell models in the human mammary epithelial cell line HMLE and the mouse mammary epithelial cell line EpRas (Supplementary Fig. 3a, c). We observed that ectopic expression of *EGFL9* had no effect on cell proliferation in either cell line (Fig. 2a, c) but showed a significant increase in cell migration and invasion in both cell lines (Fig. 2b, d; Supplementary Fig. 3b, d).

We next established two model cell lines with shRNA-mediated *EGFL9* knockdown based on the highly metastatic 4T1 and SUM159 cell lines (Supplementary Fig. 3e, g, Supplementary Table 3). No significant difference in cell proliferation was observed between these model cell lines and their vector control counterparts (Fig. 2e, g). However, knockdown of *EGFL9* expression in 4T1 cells led to a marked decrease in cell migration (***$P < 0.001$, Fig. 2f, left panel, Supplementary Fig. 3f) and invasion (***$P < 0.001$, Fig. 2f, right panel, Supplementary Fig. 3f). Similarly, knockdown of *EGFL9* expression in SUM159 cells led to significantly reduced cell migration (***$P < 0.001$, Fig. 2h, left panel, Supplementary Fig. 3h) and invasion (***$P < 0.001$, Fig. 2h, right panel, Supplementary Fig. 3h).

We next investigated whether *EGFL9* is involved in the EMT program. We did not observe expression changes in multiple epithelial and mesenchymal markers in HMLE cells upon overexpression of *EGFL9* by either western blotting or immunofluorescence assay (Supplementary Fig. 4). Moreover, no marked morphological change of HMLE cells with EGFL9 overexpression was observed in comparison with HMLE/LacZ control cells (data not shown). Furthermore, we did not observe signs of a

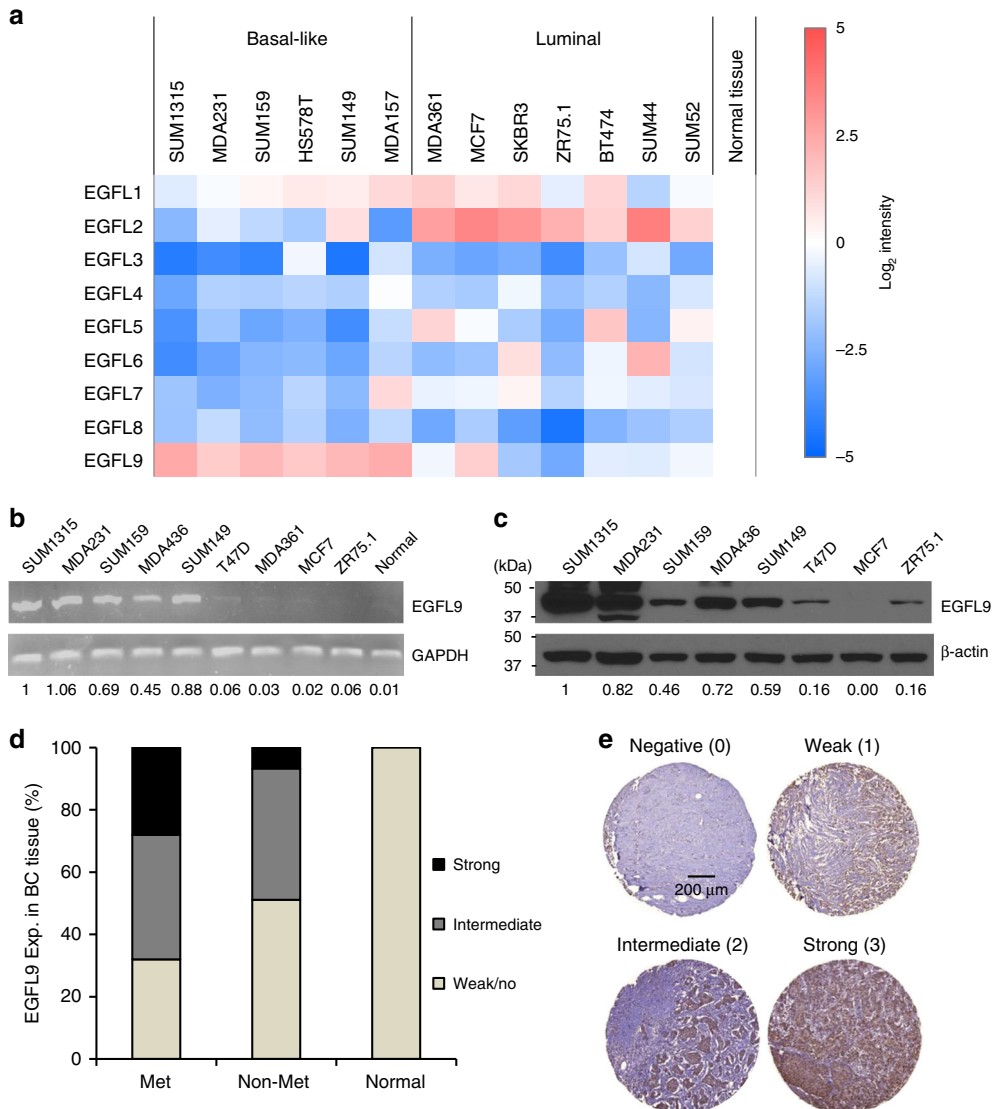

**Fig. 1** Expression of EGFL9 in breast cancer. **a** Heat map showing expression levels of the EGF-like family genes in a set of breast cancer cell lines cells. Data are normalized to GAPDH expression. Log2 intensity scale is shown on the right. **b** Expression of *EGFL9* at the RNA level in human breast cancer cell lines. The top panel shows *EGFL9* RNA expression examined by RT-PCR. The bottom digits show quantitation of the RT-PCR results. GAPDH was used as a loading control for RNA. **c** Expression of EGFL9 at the protein level in human breast cancer cell lines. The top panel shows the EGFL9 protein expression level examined by western blotting. The bottom digits show the quantitation of the EGFL9 protein expression level examined by western blot analysis. β-Actin was used as a protein loading control. **d** Summary of the EGFL9 IHC results in human breast tumor tissue microarray. **e** Expression of EGFL9 protein in human breast tumors. The panels show representative figures of the immunohistochemistry assay. 0 is no staining, 1 is an example of weak staining, 2 is intermediate staining, 3 is strong staining. Scale bar: 200 μm

mesenchymal to epithelial transition (MET) in either the 4T1 or SUM159 cell models with *EGFL9* knockdown at both the molecular and cellular level (data not shown). These data suggest that ectopic expression of *EGFL9* alone is not sufficient to trigger a complete EMT.

**EGFL9 affects breast cancer distant metastasis in vivo**. To examine the effects of the *EGFL9* gene on tumor growth and distant metastasis in vivo, EpRas cells expressing either control vector or *EGFL9* were injected into the mammary glands of nude mice. Primary tumor growth was measured, and lungs were examined for metastasis 4 weeks after cancer cell injection. We did not observe tumor growth advantage of EpRas cells with *EGFL9* overexpression (data not shown). Similarly, the tumor burden in the control vector group and the *EGFL9* overexpression group were not significantly different over the 4-week period

(Fig. 3a). Metastasis was quantified based on H&E staining of lung sections. A significant increase in the number of microscopically visible pulmonary metastases was observed upon ectopic expression of *EGFL9* (P = 0.001) (Fig. 3b). Moreover, a larger metastatic tumor area (tumor area/total lung area) was observed in the *EGFL9* overexpression cells (3–15%) compared to the control (0–2%) implanted group (P = 0.001) (Fig. 3c, d).

To validate the role of *EGFL9* in tumor growth and distant metastasis, 4T1 cells with either *EGFL9* knockdown or a non-target control were injected into the fat pads of BALB/C mice. Mice were sacrificed after 30 days of transplantation when the control group of mice reached the humane endpoint. Immediately before sacrifice, tumor size and metastatic progression were analyzed by magnetic resonance imaging (MRI). We found that tumor size was similar between these two groups (Fig. 4a, c). However, peritumor and axillary lymph node metastases were

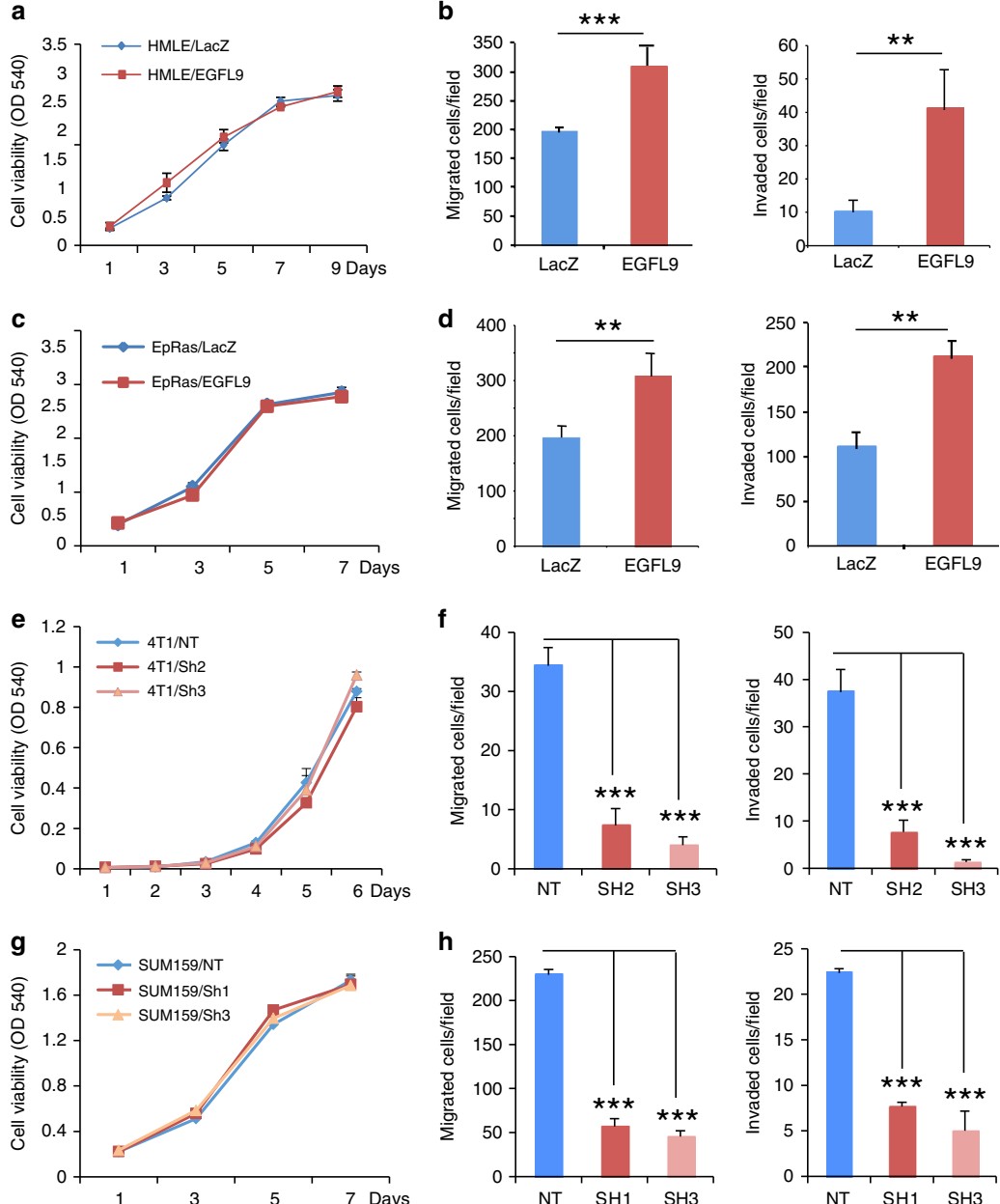

**Fig. 2** The effect of *EGFL9* on cell motility in vitro. **a** The ectopic expression of *EGFL9* does not change cell proliferation in the HMLE cell line. Cell proliferation was measured by MTT assay over 9 days. **b** The ectopic expression of *EGFL9* increased cell migration (left panel, ***P < 0.001) and invasion (right panel, **P < 0.01) in HMLE cells. **c** The ectopic expression of *EGFL9* does not change cell proliferation in the EpRas cell model. Cell proliferation was measured by MTT assay over 7 days. **d** The ectopic expression of *EGFL9* increased cell migration (left panel, **P < 0.01) and invasion (right panel, **P < 0.01) in EpRas cells. **e** Knockdown of *EGFL9* expression does not affect the cell proliferation in 4T1 cells. Cell proliferation was measured by MTT assay over 6 days. **f** Knockdown of *EGFL9* expression significantly decreased migration (left panel, shRNA2/non-target control = 21% and shRNA3/non-target control = 12%, ***P < 0.001) and invasion (right panel, shRNA2/non-target control = 19% and shRNA3/non-target control = 5.5%, ***P < 0.001) in 4T1 cells. **g** Knockdown of *EGFL9* expression does not affect proliferation of SUM159 cells. Cell proliferation was measured by MTT assay over 7 days. **h** Knockdown of *EGFL9* expression decreased migration (left panel, shRNA1/non-target control = 22.2% and shRNA3/non-target control = 21%, ***P < 0.001) and invasion ability (right panel, shRNA1/non-target control = 32% and shRNA3/non-target control = 16%, ***P < 0.001) in SUM159 cells. All experiments were performed in triplicate. The data number was the mean value from five randomly selected field views. Each bar represents the mean ± SEM (standard error of the mean). *P* values were determined by unpaired two-tailed *t*-test

detected in all five mice transplanted with 4T1/NT cells, while peritumor lymph node metastases were detected in only two of the five mice injected with 4T1/EGFL9-sh2 cells. No axillary lymph node metastases were detected in the five mice injected with 4T1/EGFL9-sh2 cells (Fig. 4a, d). Finally, four out of five mice transplanted with 4T1/NT developed lung metastases, while

only one of five mice with 4T1/EGFL9-sh2 showed a detectable metastatic lesion in the lung (Fig. 4b, d).

Our MRI findings were confirmed by pathological analysis of tumor sections. Final tumor burden was not significantly different between the 4T1 cells with vector control and *EGFL9* knockdown groups (data not shown). However, we observed a 50% decrease

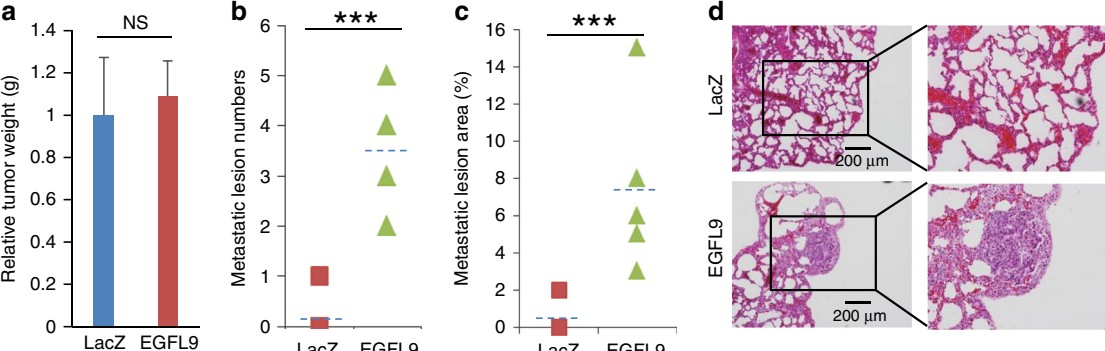

**Fig. 3** Overexpression of *EGFL9* promotes distant tumor metastasis in vivo. **a** Overexpression of *EGFL9* in EpRas cells did not change tumor weight. EpRas/LacZ cells serve as a control model (NS, *P* = 0.59). **b** Quantification of metastatic lesions in two groups of tumors derived from EpRas/LacZ and EpRas/EGFL9 cells implanted in NCR nu/nu mice. Average metastatic lesion numbers from five serial sections in each tumor were compared. ***P* < 0.001. *P* values were determined by unpaired two-tailed *t*-test. **c** Quantification of metastasis by lesion area in two groups of tumors derived from EpRas/LacZ and EpRas/EGFL9 cells implanted in NCR nu/nu mice. Relative metastatic lesion areas (tumor lesion/lung section) from sections with the highest metastatic lesion number in each tumor were compared. ***P* < 0.001. *P* values were determined by unpaired two-tailed *t*-test. **d** Representative H&E staining pictures of lung sections from mice. IHC results showed significant metastatic lesions in *EGFL9* transfected EpRas cells (low panel) in comparison with EpRas transfected with LacZ (top panels). Magnification: ×40 (left panels) and ×400 (right panels). Scale bar: 200 μm

in the number of microscopically visible pulmonary metastases following knockdown of *EGFL9* (*P* < 0.001) (Fig. 4e). Moreover, a smaller tumor area (tumor area/total lung area) was observed in the *EGFL9* knockdown group (0–7%) compared to the non-target shRNA-transfected cell group (5–25%) (*P* < 0.001) (Fig. 4f, g). Based on these results, we concluded that 4T1 cell metastasis in mice was significantly inhibited after knockdown of EGFL9.

**EGFL9 specifically activates the cMET signaling pathway.** To explore the downstream signaling pathways controlled by EGFL9, we compared the phosphorylation of 49 receptor tyrosine kinases in HMLE/EGFL9 to the control HMLE/LacZ cells using a Human Phospho-Receptor Tyrosine Kinase Array. After normalization, the phosphorylation level of cMET and EGFR were elevated by 5- and 1.3-fold in HMLE/EGFL9 cells, respectively, as compared to HMLE/LacZ control cells (Fig. 5a, b), suggesting overexpression of EGFL9 specifically activates cMET signaling. This result was confirmed by western blotting, which showed that cMET and EGFR phosphorylation were increased due to expression of EGFL9, while knockdown of EGFL9 decreased cMET and EGFR phosphorylation (Fig. 5c). Several critical intracellular signaling molecules, including FAK, AKT, and ERK, showed concomitant phosphorylation upon activation of cMET and EGFR. Similarly, the activation of these downstream kinases was inhibited when EGFL9 was knocked down (Fig. 5c). EGFL9-dependent cMET and ERK phosphorylation was confirmed by immunofluorescence assay in both HMLE/EGFL9 and SUM159 cell models (Supplementary Fig. 5a, b). These results demonstrate that EGFL9 expression increases cMET phosphorylation and activates cMET downstream targets.

**Inhibition of cMET reverses EGFL9 induced cell signaling.** To investigate whether cMET is the mediator of EGFL9 signaling, we treated HMLE/EGFL9 cells with JNJ38877605, a specific inhibitor of cMET. We found that treatment with JNJ38877605 for 24 h effectively blocked the phosphorylation of cMET in a dose-dependent manner (Fig. 5d). Moreover, we observed a concomitant decrease in p-EGFR, p-FAK, pAKT, and pERK along with the decrease of the cMET phosphorylation (Fig. 5d). Additionally, JNJ38877605 treatment effectively reversed EGFL9-driven cell migration and invasion in HMLE/EGFL9 cells (Fig. 5e). Consistent with the above results, we found that treating

SUM159 cells with JNJ38877605 caused similar signaling inhibition (Fig. 5f). Together, these data strongly suggest that the cMET signaling cascade is an important effector for mediating the EGFL9 effect on cell invasiveness.

To confirm that cMET is a critical mediator required for EGFL9 to activate downstream signaling, HMLE cells was treated with JNJ38877605, for inhibition of cMET activation, and EGFL9 expression was sequentially induced with EGFL9-expressing lentivirus. Western blot analysis showed that phosphorylation of EGFR and AKT could not be induced by EGFL9 in cells pretreated with cMET inhibitor JNJ38877605 (Supplementary Fig. 5c), suggesting that cMET activation is required for EGFL9 to activate these metastasis-related signaling pathways.

**EGFL9 physically interacts and co-localizes with cMET.** Structural analysis of EGFL9 revealed that EGFL9 contains a signal peptide and multiple disulfide bonds at the N-terminus and a transmembrane domain at the C-terminus, suggesting EGFL9 is possibly a membrane-bound protein. To interrogate the potential interaction between EGFL9 and cMET, protein extracts from HMLE/EGFL9 cells were immunoprecipitated (IP) with anti-V5 antibody followed by immunoblotting (IB) with antibodies against cMET. The results confirmed an interaction between the overexpressed EGFL9 with endogenous cMET (Fig. 6a, left panel). Reciprocally, IP with antibodies against endogenous cMET followed by immunoblotting with anti-V5 antibody against overexpressed EGFL9-v5 showed that EGFL9 was efficiently immunoprecipitated by cMET (Fig. 6a, right panel). Next, total protein from SUM159 cells was extracted and co-immunoprecipitation experiments were performed with antibodies detecting both endogenous proteins. Immunoprecipitation with anti-cMET antibody followed by immunoblotting with anti-EGFL9 antibody demonstrated that the EGFL9 protein efficiently co-immunoprecipitated with cMET (Fig. 6b). These results support the conclusion that EGFL9 is physically associated with cMET in breast cancer cells.

To further study the co-localization of EGFL9 and cMET at the subcellular level, we performed immunofluorescence and confocal imaging in HMLE/EGFL9 cells. The confocal images show localization of both EGFL9 and cMET on the cell membrane along with strong staining in the perinuclear region (Fig. 6c, Supplementary Movie 1). The co-localization of these two proteins was evaluated with Volocity software, which confirmed

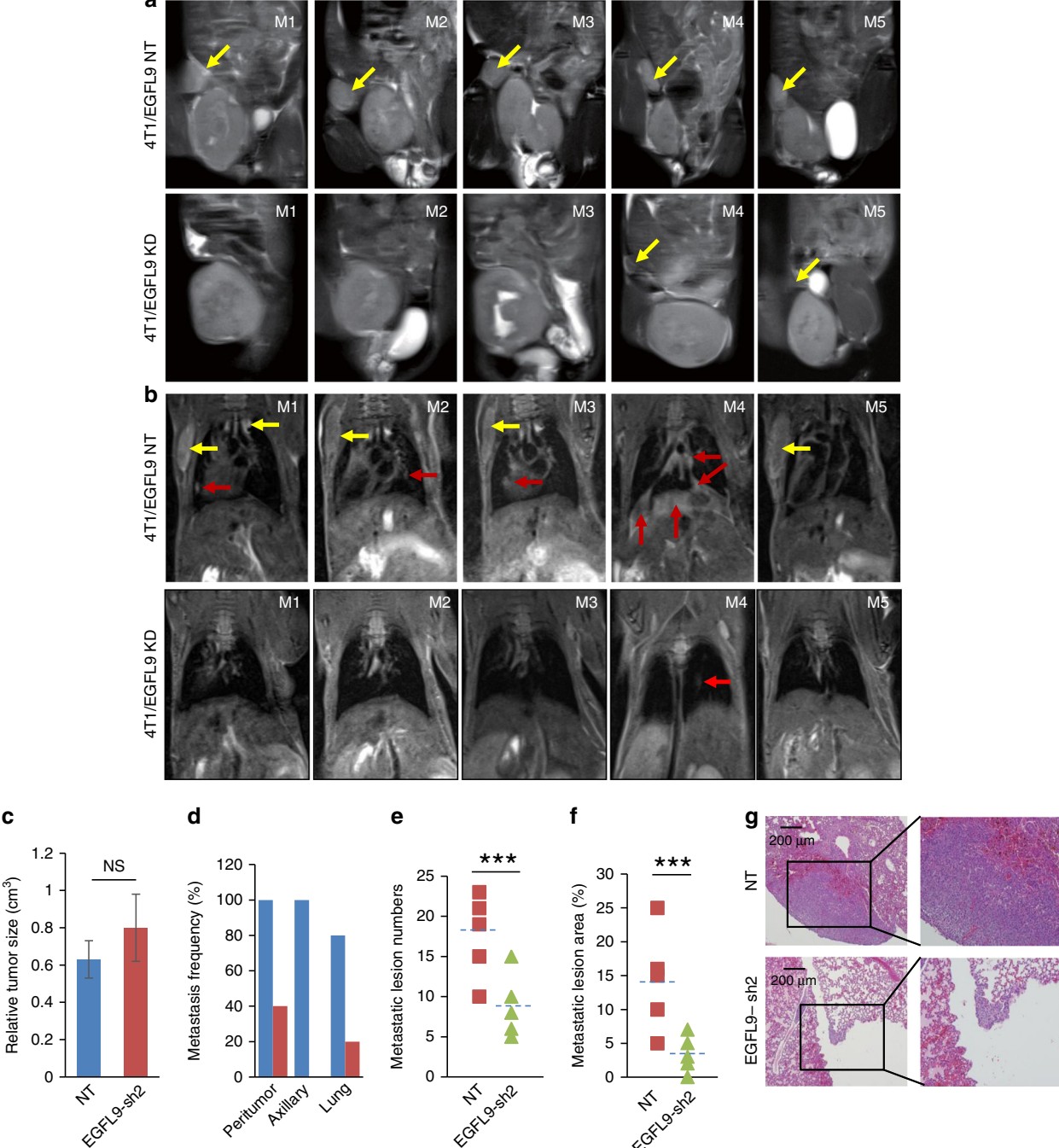

**Fig. 4** Knockdown of *EGFL9* inhibits distant tumor metastasis in vivo. **a** MRI examination identified peritumor lymph nodes metastasis in 4T1/NT injected mice (bottom panels), but not in 4T1/EGFL9-sh2 injected mice (top panels). Five cases of 4T1 tumor xenograft mouse model were shown. Yellow arrow showed the peritumor localization in each case. **b** MRI examination identified axillary lymph nodes and lung metastasis in 4T1/EGFL9-NT-injected mice (bottom panels), but not in 4T1/EGFL9-sh2-injected mice (top panels). Five cases of 4T1 tumor xenograft mouse model were shown. Yellow arrow shows the localization of axillary lymph nodes metastasis and red arrow showed the lung metastasis in each case. **c** Quantitative tumor size summary by MRI detection. No significant difference was observed in two groups of tumors (NS, *P* > 0.05). Each bar represents the mean ± SEM (standard error of the mean, *n* = 5). *P* value was determined by unpaired two-tailed *t*-test. **d** The frequencies of metastasis identified by MRI in peritumor lymph nodes (5/5 vs 2/5), axillary lymph nodes (5/5 vs 0/5), and lungs (4/5 vs 1/5) of BALB/c mice (*n* = 5 in each group) with control 4T1 (blue bar) and 4T1/EGFL9-sh2 (red bar) cells. **e** Comparison of metastatic lesion numbers in two groups of tumors. Average metastatic lesion numbers from five series sections were counted for each tumor. ***P* < 0.001. **f** Comparison of metastasis lesion area in two groups of tumors. Relative metastatic lesion areas (tumor lesion/lung section) from sections with highest metastatic lesion number in each tumor were compared. ***P* < 0.001. For both **e** and **f** panels, *P* values were determined by unpaired two-tailed *t*-test (*n* = 5). **g** Representative H&E staining of lung sections from mice. IHC demonstrated significant metastatic lesions in NT (non-targeting shRNA)-transfected 4T1 cells (top panels) in comparison with EGFL9-sh2-transfected 4T1 (bottom panels). Magnification: ×40 (left panels) and ×100 (right panels). Scale bar: 200 μm

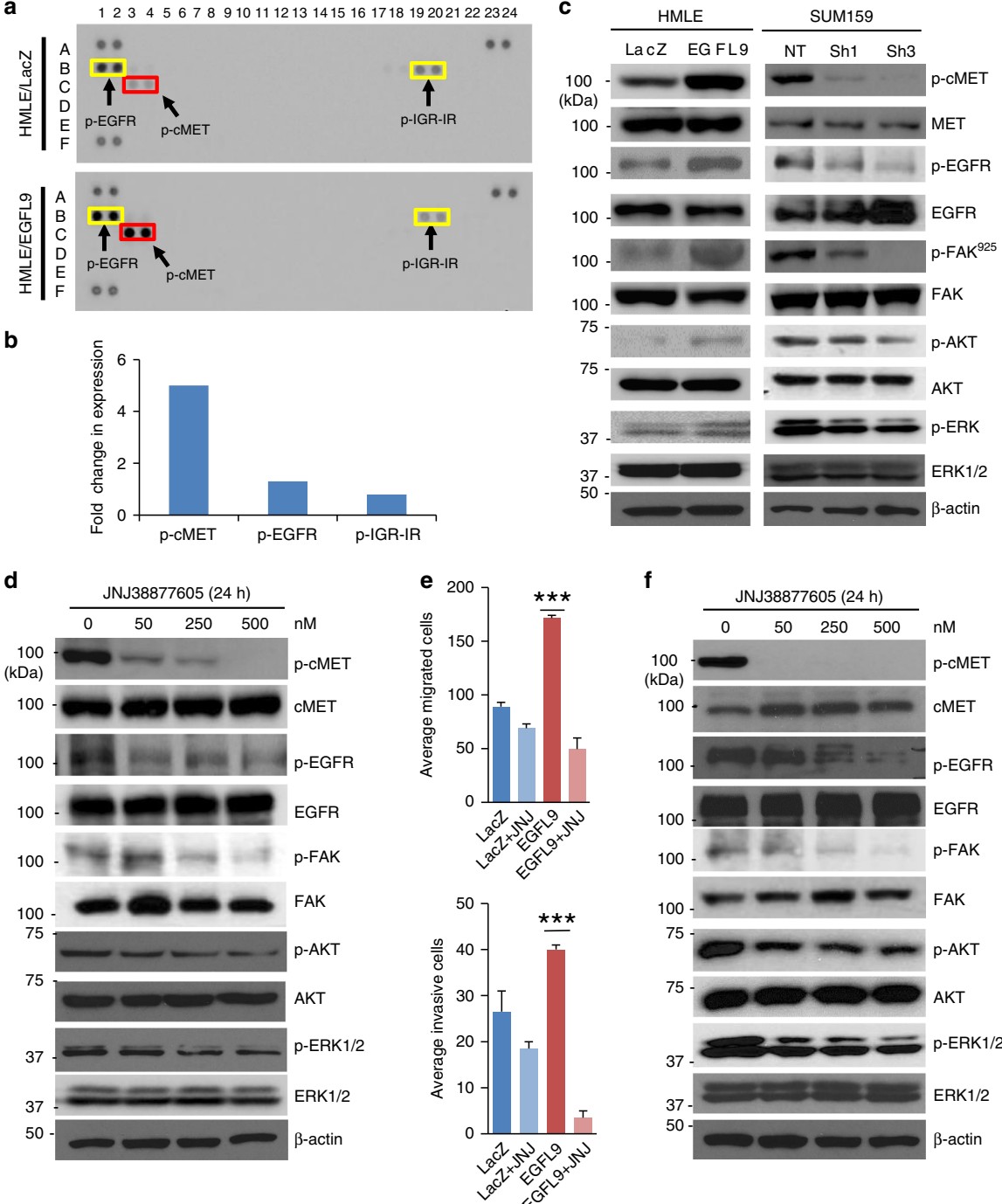

**Fig. 5** EGFL9 specifically activates cMET regulated signaling pathways. **a** Monitoring the receptor tyrosine kinase activation using a phospho-receptor tyrosine kinase array. Two isogenic cell lines, HMLE/LacZ and HMLE/EGFL9, were analyzed for signaling activation of a set of 49 receptor tyrosine kinases. **b** Quantitation of activation of cMET, EGFR, and IGR-IR in HMLE/EGFL9 cells in comparison with HMLE/LacZ. **c** Western blotting analysis of ectopic expression of EGFL9 in HMLE cells (left panel) and knockdown of EGFL9 in SUM159 cells. Phosphorylation and total cMET and cMET downstream targets (EGFR, FAK, AKT, and ERK) were examined. **d** Western blotting of cMET and its downstream targets in HMLE/EGFL9 cells after treatment with various concentrations of JNJ38877605 for 24 h. The antibodies used for the western blotting analysis were listed in the panels. **e** The treatment with JNJ38877605 leads to a significant decrease of cell migration (top panel, ***$P < 0.001$), invasion (bottom panel, ***$P < 0.001$) in HMLE/EGFL9, but not in HMLE/LacZ cells. Each bar represents the mean ± SEM (standard error of the mean). For three independent experiments, $P$ value was determined by unpaired two-tailed $t$-test. **f** Western blotting of cMET and its downstream targets in SUM159 cells after treatment with various concentrations of JNJ38877605 for 24 h. The antibodies used for the western blotting analysis were listed in the panels

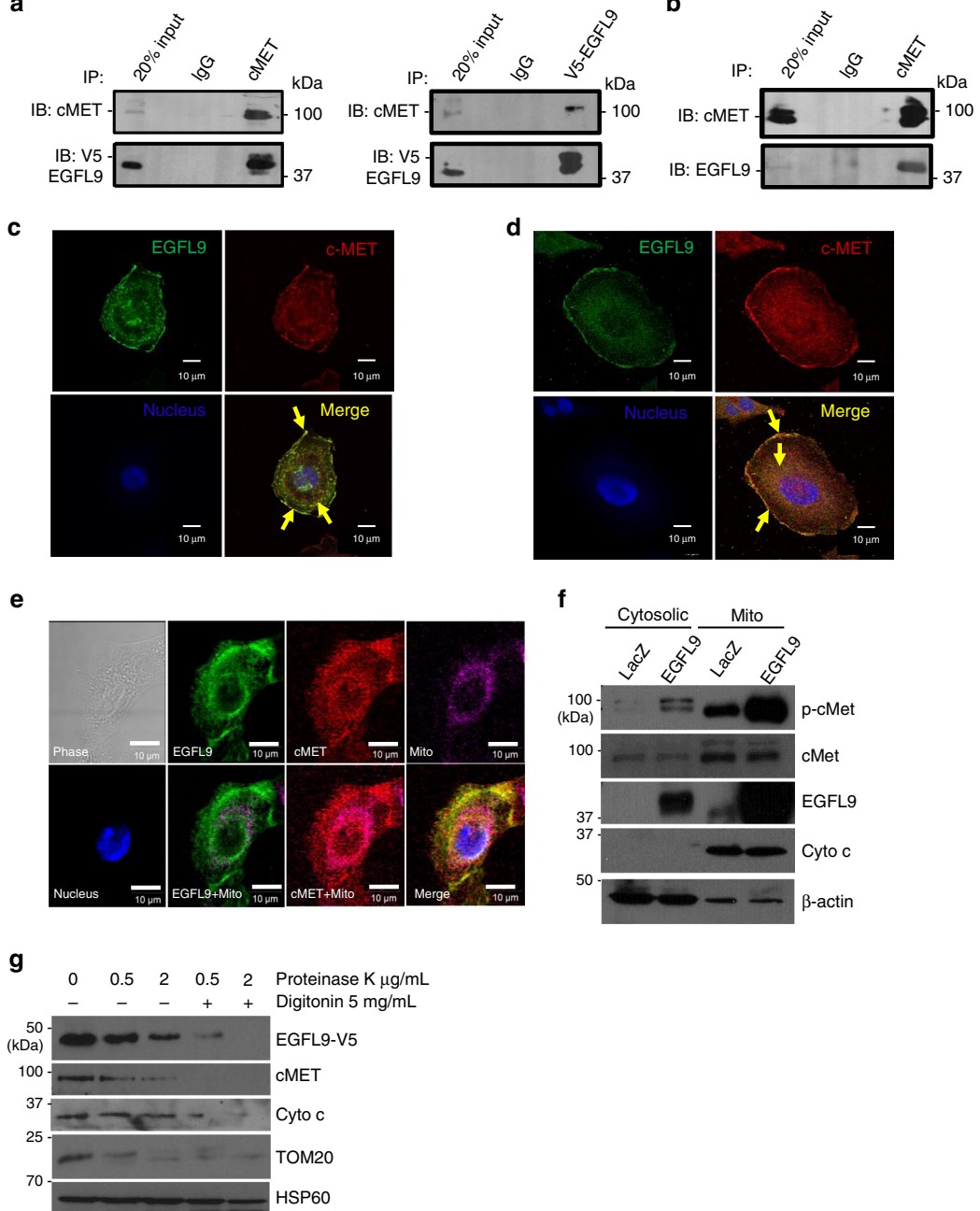

**Fig. 6** Physical association and co-localization of EGFL9 and cMET. **a** Interaction of EGFL9 and cMET in HMLE/EGFL9 cells. Left panel: immunoprecipitation was performed with anti-v5 antibody followed by immunoblotting with cMET antibody. Right panel: immunoprecipitation was performed with anti-cMET followed by immunoblotting with anti-V5 antibody. **b** Interaction of EGFL9 and cMET in SUM159 cells. Immunoprecipitation was performed with anti-cMET antibody followed by immunoblotting with anti-EGFL9 antibody. For both **a** and **b**, IgG was used as the IP-control. **c**, **d** Subcellular co-localization of EGFL9 and cMET in HMLE/EGFL9 (**c**) and SUM159 (**d**) cells. The distribution of ectopic expressed EGFL9 was detected with anti-v5 antibody (**c**). Endogenous EGFL9 was detected by EGFL9 antibody (**d**). cMET was detected with anti-cMET antibody (**c**, **d**). DAPI staining was included to visualize the cell nucleus. Scale bar: 10 μm. **e** Co-localization of EGFL9 and cMET in mitochondria in HMLE/EGFL9 cells. Mitochondria were stained with Mitotracker deep-red dye. The distribution of ectopic expressed EGFL9 was detected with an anti-V5 antibody. cMET was detected with an anti-cMET antibody. DAPI staining was included to visualize the cell nucleus. Scale bar: 10 μm. **f** Cytosolic and mitochondrial fractions were isolated from HMLE/EGFL9 cells and subjected to SDS-PAGE followed by probing with specific antibodies as indicated. Cyt c, mitochondrial marker; β-actin, cytosolic marker. **g** Proteinase K protection assay. Equal amounts of purified mitochondria were treated as indicated. Lysed protein was separated by SDS-PAGE and immunoblotted using the indicated antibodies

that the whole cell and cell membrane co-localization levels were significant when measured by Pearson's correlation and co-localization coefficient *M* values (Supplementary Fig. 6a–c). This co-localization pattern was confirmed by immunofluorescence and confocal imaging in the SUM159 cells by detection of

endogenous EGFL9 and cMET proteins (Fig. 6d, Supplementary Fig. 6d–f).

Next, fragment mapping was performed to probe the physical association between EGFL9 and cMET in detail. Western blotting did not detect EGFL9 protein expression when the EGF-rich

domain in EGFL9 was deleted (data not shown), suggesting that the EGF-rich domain in EGFL9 is essential for the stability of the EGFL9 protein. Next, we generated a cMET mutant with a deletion of the SEMA domain (AA 27–515), which has been shown to be responsible for receptor dimerization (Supplementary Fig. 7a)[17]. SEMA deletion does not alter the localization of cMET protein (Supplementary Fig. 7b). We performed a V5-IP of EGFL9 in the presence of co-transfected wild-type cMET or mutant cMET (cMETdel). The results of the co-IP western blot demonstrate that mutant cMET (cMETdel) is still able to interact with EGFL9 (Supplementary Fig. 7c), suggesting that receptor dimerization of cMET is not required for its interaction with EGFL9.

**The EGFL9–cMET complex localizes to the mitochondria.** cMET has been reported to be expressed on the cell surface as a receptor tyrosine kinase[18]. While one previous report identified cMET localization in the mitochondria of cancer cells via quantitative proteomics, this finding was not validated or functionally evaluated[19]. The perinuclear localization of the EGFL9-cMET complex prompted us to explore whether the complex co-localizes with mitochondria. Using immunofluorescence and confocal microscopy, we observed that both EGFL9 and cMET co-localizes with the mitochondria around the nucleus in both HMLE/EGFL9 and SUM159 cells (Fig. 6e, Supplementary Figs. 6g, h, and 8a, b, and Supplementary Movie 2). We also analyzed the organelle fractions from both HMLE/EGFL9 and SUM159 cells. The preferential accumulation of p-cMET and EGFL9 in mitochondrial fractions over cytoplasmic fractions in both cell models was detected by western blotting. The total cMET protein showed similar distribution in mitochondria and cytoplasmic fractions. Western blots were also probed for mitochondria marker cytochrome *c*, which showed specific expression in mitochondria (Fig. 6f, Supplementary Fig. 8c).

To further determine the mitochondrial localization of EGFL9 and cMET, we performed a proteinase K protection assay. We observed that EGFL9 and cMET proteins were protected against degradation by 0.2–2 mg/ml of proteinase K but underwent degradation when subjected to combined treatment with proteinase K and digitonin. Cytochrome *c* (Cyt *c*), an intermembrane space protein, showed a similar pattern with these treatments. Tom20, an outer membrane protein, was digested by proteinase K, while Hsp60, a matrix protein, was protected against proteinase K or proteinase K plus digitonin treatment. These results indicate that EGFL9 and cMET are localized within the mitochondria and associate with the inner membrane (Fig. 6g).

**EGFL9 regulates cellular metabolism.** To identify mediators of EGFL9 activity in mitochondria, we purified the mitochondria from HMLE/EGFL9 cells and performed EGFL9 IP mass spectrometry. We identified 23 mitochondrial proteins were significantly enriched (data not shown). Of interest, we found COA3, a COX assembly factor, as a potential interacting partner of EGFL9 within the mitochondria. To functionally validate EGFL9 localization in the mitochondria, we examined EGFL9 and COA3 interaction with a Bimolecular Fluorescence Complementation (BiFC) assay[20,21]. The results showed that only cells co-transfected with split-Venus EGFL9 and COA3 showed specific reconstituted yellow fluorescence protein (YFP) signals (Fig. 7a, Supplementary Fig. 9, and Supplementary Movie 3). Specific YFP signal was not identified in cells transfected with only COA3-VC (Fig. 7a) or EGFL9-V5 (data not shown). The interaction of EGFL9 and COA3 was confirmed by a co-IP western blotting (Fig. 7b). COA3 is a protein localized at the inner

mitochondria membrane and recently was shown to stabilize cytochrome *c* oxidase 1 (COX1) and promote COX assembly in human mitochondria[22,23]. The interaction between EGFL9 and COA3 suggests that EGFL9 may regulate mitochondria function through disrupting COX assembly.

Exploring this hypothesis further, we found that the activity of COX was decreased by 54.5% in the EGFL9 overexpression cell model compared to LacZ control (Fig. 7c), indicating that the electron transport chain (ETC) was targeted by EGFL9 signaling. These changes in COX activity translated to the intact cell level; oxygen consumption rate (OCR), measured with a Seahorse XF Analyzer, showed that basal respiration was reduced by 68% in HMLE/EGFL9 cells as compared to control (Fig. 7d). Furthermore, the maximal respiration rate (after addition of FCCP) was also significantly lower in HMLE/EGFL9 cells (Fig. 7d). To investigate the mechanism of loss of COX function, COX complex was immunoprecipitated with anti-COX IV antibody and immunoblotted with COX II and phosphotyrosine antibodies. The results show that COX phosphorylation was significantly increased in cells with EGFL9 overexpression over cells with LacZ control (Fig. 7e). Consistent with these results, we observed a significant increase of lactate production (1.58-fold) (Fig. 7f) and glucose uptake (1.51-fold) (Fig. 7g) in HMLE/EGFL9 cells over HMLE/LacZ cells, indicating a bioenergetic conversion from an oxidative to a glycolytic state. In addition, knockdown of EGFL9 in SUM159 cells decreased lactate production (−40.8% and −27.1%) and glucose uptake (−80% and −40%) (Supplementary Fig. 8d, e). Moreover, when HMLE/EGFL9 cells were treated with the cMET inhibitor JNJ38877605, EGFL9 subcellular distribution and expression were not changed along with inhibition of cMET phosphorylation. Instead, we observed a modest 23% increase in COX activity, a 28% decrease in lactate production, and no significant change in glucose uptake (Supplementary Fig. 10), suggesting EGFL9-promoted metabolic changes are only partially dependent on cMET activity.

We next examined the mitochondrial membrane potential in HMLE/EGFL9 and HMLE/LacZ cells using FACS analysis. We detected significantly increased accumulation of the potentiometric probe JC-1 in HMLE/EGFL9 cells compared to HMLE/LacZ cells (Supplementary Fig. 11a, b), suggesting mitochondrial hyperpolarization. This is also consistent with increased total cellular ROS (1.37-fold increase) in HMLE/EGFL9 cells over HMLE/LacZ cells (Fig. 7h). The production of superoxide by mitochondria was measured using the probe MitoSOX™, and a 1.46-fold increase was found in HMLE/EGFL9 mitochondria compared to HMLE/LacZ mitochondria (Fig. 7i). We then measured the ATP generation in cells with or without EGFL9 overexpression and identified a 40% increase in ATP generation in cells with EGFL9 expression when compared to control cell model (Supplementary Fig. 11c), despite apparent dysfunction of COX activity and OXPHOS impairment. This result is similar to a phenomenon frequently observed in cancer in which glycolysis is upregulated and glycolytic ATP is increased[24]. To explore whether ATP production in EGFL9 expression cells is mainly contributed by glycolysis, we measured ATP level after blocking OXPHOS using two mitochondria respiratory chain inhibitors oligomycin and rotenone. Our results showed that ATP production maintained at high level after OXPHOS inhibition in HMLE/EGFL9 cells in comparison with control HMLE/LacZ cells (Supplementary Fig. 11d), suggesting a great portion of ATP production is attributed to glycolysis in cells with ectopic expression of EGFL9.

**Targeting cMET and glycolysis reverses cancer stemness.** Given the observation that cancer stemness is associated with cancer

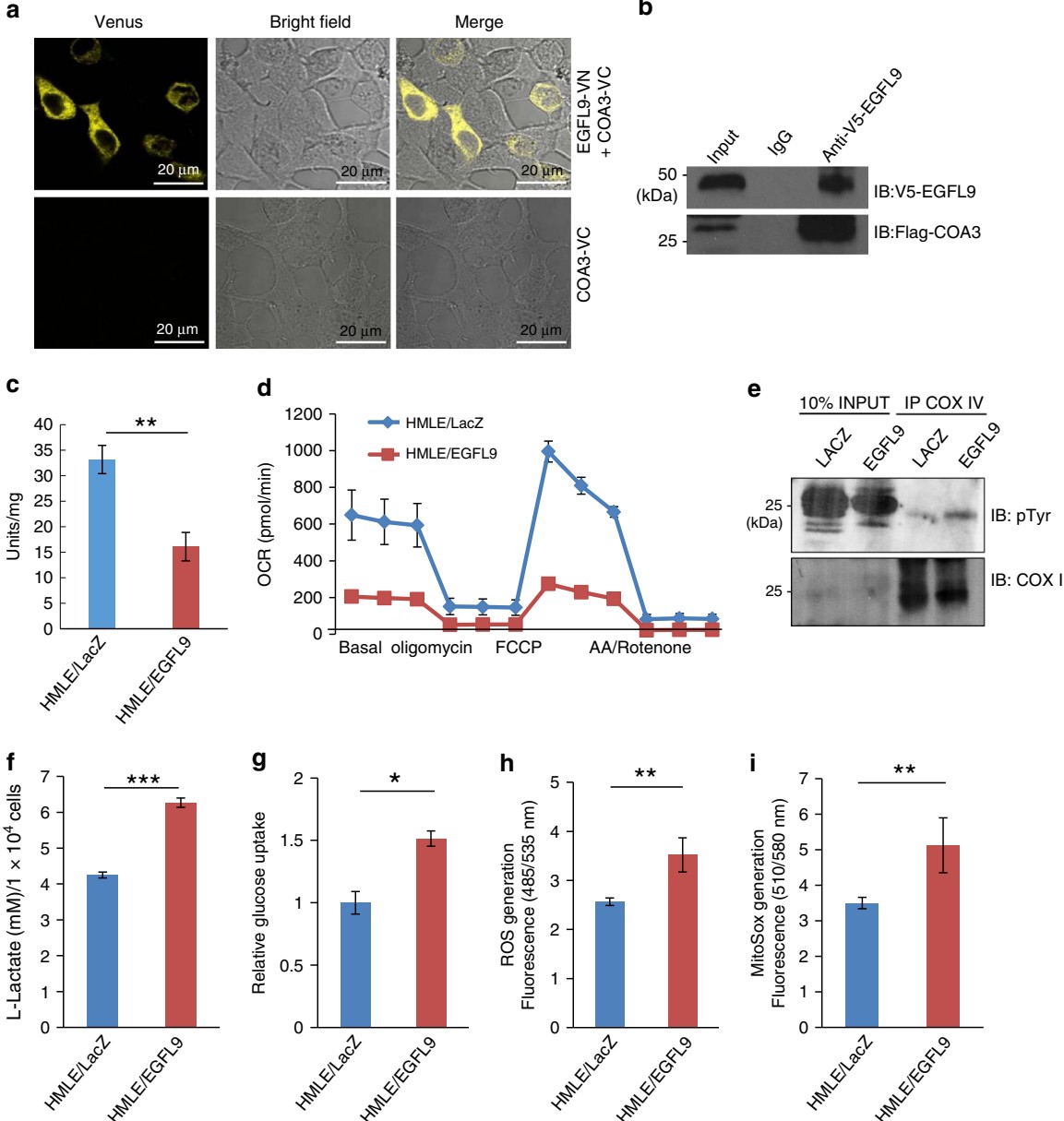

**Fig. 7** EGFL9 regulates cellular metabolism. **a** Visualization of EGFL9-COA3 interaction in vivo by BiFC live-cell imaging performed in 293 T cells. The top panels show 293 T cells transfected with p3xFlag-CMV-EGFL9-VN and p3xFlag-CMV-COA3-VC. The bottom panels show 293 T cells transfected with COA3-VC alone. Images of Venus (YFP), bright filed, and merge are shown. Scale bar: 20 μm. **b** BiFC result was confirmed by Co-IP western blot analysis. pLent-6-EGFL9-V5 and p3xFlag-CMV-COA3 constructs were co-transfected in 293 T cells. Co-IP was performed with anti-V5. Western blotting was then performed with anti-Flag antibody. **c** COX specific activity was measured using a Cytochrome Oxidase Activity Colorimetric Assay Kit. In all, 2.5 μg of mitochondria were mixed with reduced cytochrome oxidase and buffer per the manufacturer's instructions. The OD at 550 nm was measured and the maximum linear decay was used to calculate activity (units/mg). Each bar represents the mean ± SEM (standard error of the mean of a representative experiment performed in triplicate). $P$ value was determined by unpaired two-tailed $t$-test. **$P = 0.03$. **d** Basic oxygen consumption rate (OCR) and maximum oxygen consumption rate were determined in a Seahorse bioanalyzer after normalization to cell numbers. FCCP, a potent uncoupler of mitochondrial oxidative phosphorylation, was added to both cell culture experiments. Several inhibitors of mitochondrial respiratory chain complex including Oligomycin, Rotenone, and Antimycin were also added. **e** Phosphorylation of COX was significantly increased in HMLE/EGFL9 cells in comparison with HMLE/LacZ cells. Mitochondrial protein was subject to IP with COX IV antibody and then blotted with COX II and pTyr antibodies. **f** Lactate levels in HMLE/LacZ and HMLE/EGFL9 culture medium were measured using a glycolysis cell-based assay. **g** The glucose concentration in HMLE/LacZ and HMLE/EGFL9 culture medium was measured with the Amplex Red Glucose Assay Kit. **h**, **i** Whole-cell reactive oxygen species (ROS) (**h**) and mitochondrial superoxide (**i**) were measured in HMLE/LacZ and HMLE/EGFL9 cells. For all experiments from **f** to **i**, each bar represents the mean ± SEM (standard error of the mean of a representative experiment performed in triplicate). $P$ values were determined by unpaired two-tailed $t$-test. *$P < 0.05$; **$P < 0.01$, and ***$P < 0.001$

metastasis and metabolic reprogramming[25,26], we next investigated whether EGFL9 expression leads to enrichment of the stem cell subpopulation. ALDH$^+$ and CD44$^+$/CD24$^-$ are two major CSC subpopulations identified in human breast cancer cells. These CSC populations are distinct in both morphology and function[27]. We found that the ALDH$^+$, but not CD44$^+$/CD24$^-$, subpopulation was enriched by ectopic expression of EGFL9 (Fig. 8a). This enrichment could be reversed not only by the cMET inhibitor JNJ38877605, but also by the glycolysis inhibitor 2-DG in a dose-dependent manner (Fig. 8a–d). Moreover, we observed a concomitant increase in tumorsphere formation upon ectopic EGFL9 expression. JNJ38877605 or 2-DG treatment independently reversed this capability, while the combination of JNJ38877605 and 2-DG showed a more profound effect in decreasing mammosphere formation (Fig. 8e). The results obtained for 2-DG were reproduced using another glycolysis inhibitor NHI-2, which specifically targets lactate dehydrogenase-A (LDH-A, LDHA), a key enzyme necessary to sustain glycolysis. Both NHI-2 treatment alone (Supplementary Fig. 12a–c) and combinational treatment of NHI-2 and JNJ38877605 (Supplementary Fig. 12d) reduced mammosphere formation in a similar manner as 2-DG.

In line with the above observation, ectopic expression of EGFL9 promoted resistance of mammary epithelial cells to two first line chemotherapeutic drugs Doxorubicin (Dox) and Paclitaxel (Pac) (Fig. 8f, g). We found that neither JNJ38877605 nor 2-DG treatment alone sensitized EGFL9 overexpression cells to the chemotherapeutic agents Dox or Pac (data not shown), but the combination of these two drugs significantly sensitized both HMLE/EGFL9 (Fig. 8h, i) and SUM159 cells (Fig. 8j, k) to chemotherapeutic agents Dox and Pac. Taken together, these results are consistent with our previous finding that cMET activation is only partially responsible for EGFL9-promoted metabolic changes, suggesting that cMET activation and mitochondrial dysfunction cooperatively contribute to EGFL9-driven cancer stem cell phenotype.

Finally, we examined the expression of embryonic stem cell (ESC)-associated genes with a RT-PCR stem cell array in HMLE/EGFL9 cells (Supplementary Table 4). The core of this array is formed by a set of stem cell molecular markers and regulatory genes involved in proliferation, self-renewal, and pluripotency, but the signature also contains several major components in Hedgehog, Notch, Wnt signaling pathways, which are known to be related to stemness[28]. Interestingly, only a few genes exhibited altered expression upon EGFL9 expression. Among these altered genes, we found that the expression level of pluripotency markers NANOG, OCT4, and KLF4 were increased in the EGFL9 overexpression HMLE cell model (~1.5–2.5-fold; Supplementary Fig. 13a), which is consistent with previous reports and supports the role of cMET activation in cell stemness[29]. These results were validated by western blotting and immunofluorescence (Supplementary Fig. 13b, c), suggesting that EGFL9 enhances stemness of mammary epithelial cells partially through activation of the pluripotency program.

## Discussion

EGFL9, also known as DLK2, has been intensively studied for its potential biological function in conjunction with its homolog DLK1. DLK1 may potentiate or inhibit adipogenesis depending upon the cellular context involving Notch1 expression and activation. As a DLK1 homolog, DLK2/EGFL9 may regulate adipogenesis in an opposing manner to DLK1 (refs. [30,31]). Additionally, EGFL9/DLK2, along with several other EGF-Like proteins (EGFL2, EGFL3, EGFL5, EGFL6, EGFL7, EGFL8), is enriched in the local bone environment, suggesting a potential role of EGFL9 in angiogenesis[8,32]. So far, no reports have linked EGFL9 with cancer progression and metastasis. Our results revealed that EGFL9 is preferentially expressed in TNBC and can promote breast cancer metastasis through two partially distinct mechanisms of action: activation of cMET signaling and significant metabolic reprogramming/bioenergetic conversion (Fig. 9).

One previous study suggested that cMET signaling is critical for cell cycle progression in primary effusion lymphoma[33]. However, we observed that EGFL9, which specifically activates cMET, regulates metastasis but not primary tumor growth of TNBC. This phenomenon could be due to tissue-specific cMET activation in cells or tumor microenvironment. Additionally, several recent studies have demonstrated that EMT is not required for metastasis[34,35]. There are two major cancer stem-like cell subpopulations in human breast cancer: ALDH$^+$ and CD44$^+$/CD24$^-$. The results from our research group and others show that these two subpopulations are not identical[27,36]. The ALDH$^+$ stem cell subpopulation is proliferative and remains in an epithelial-like state, while the CD44$^+$/CD24$^-$ subpopulation is quiescent, invasive, and mesenchymal-like state. In this study we observe that EGFL9 expression results in enrichment of ALDH$^+$, not the CD44$^+$/CD24$^-$, stem-like cell subpopulation, suggesting that EGFL9 enhances stemness and metastasis through an EMT-independent manner.

The relationship between metastasis and metabolic status is complicated and context dependent. Evidence in recent years has suggested that metabolic reprogramming and the Warburg effect promote metastasis[7,37,38]. In addition, mitochondria dysfunction was demonstrated to be closely correlated with tumor progression and metastasis[6,39]. Our study provides evidence indicating that EGFL9 can simultaneously promote metabolic reprogramming and cMET activation. We demonstrated that EGFL9 expression increased the abundance of the ALDH$^+$ stem cell population, which is highly stem-like and proliferative. Treatment with glycolysis inhibitors attenuated EGFL9-promoted ALDH$^+$ subpopulation enrichment and mammosphere formation. Therefore, our data agree with recent discoveries by demonstrating that metabolic reprogramming or bioenergetic switch contributes to EGFL9-driven metastatic potential through maintaining the stemness in cells with EGFL9 overexpression.

COX consumes >90% of cellular oxygen and catalyzes the terminal and rate-limiting step of the ETC[40,41], thus controlling overall flux and making it an ideal target for switching to Warburg metabolism. Loss of COX function disrupts respiratory super-complexes that are thought to have an important role in the regulation of electron transport, oxidative phosphorylation (OXPHOS), and attenuation of ROS production[40–43]. Several oncogenes including EGFR and HER2 have been reported to translocate to mitochondria and target COX via post-translational modification, resulting in COX inhibition[44–46]. Our study provided two lines of evidence to support the function of EGFL9 in regulating COX activity. First, EGFL9 expression leads to phosphorylation of COX proteins. This might occur through EGFL9-mediated activation of cMET and downstream signaling. Second, we showed that EGFL9 interacted with a COX assembly factor, COA3. This interaction may disrupt COX assembly through induced instability of COX proteins. As a future direction, we will investigate the interplay of these two mechanisms, and investigate other EGFL9 binding partners within the mitochondria.

Our study showed EGFL9 interacts with cMET in TNBC cell lines. However, how the interaction of EGFL9-cMET contributes to cancer metastasis remained unclear. We speculate that the EGFL9-cMET interaction facilitates cMET activation as a major factor to promote cancer metastasis. This hypothesis is based on two observations. First, we showed specific activation of cMET

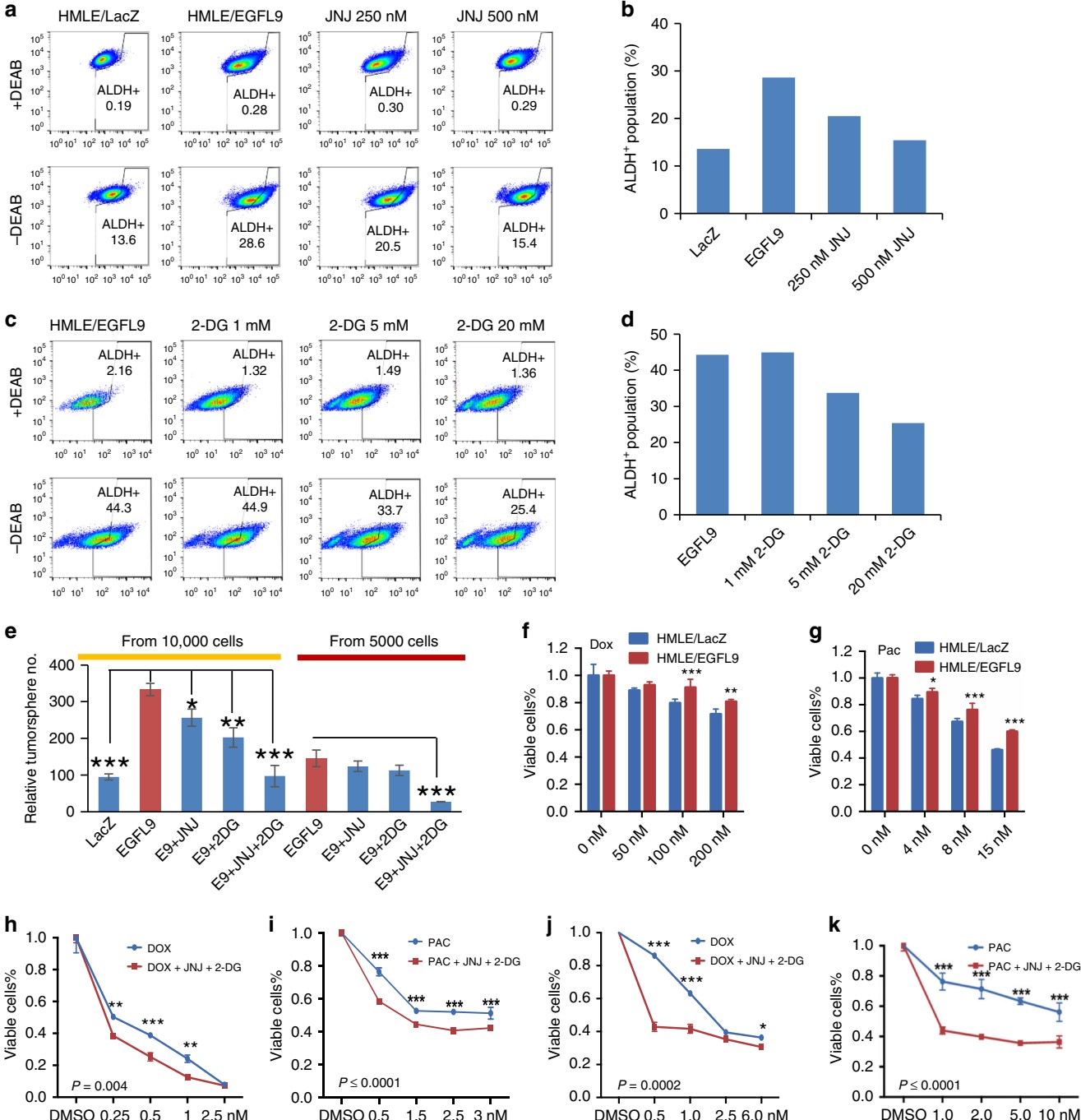

**Fig. 8** EGFL9 induces stemness in mammary epithelial cell. **a** ALDH+ FACS profiles are shown for HMLE/LacZ cells, and HMLE/EGFL9 cells with different doses of JNJ38877605 treatment. **b** Summary of percentage of ALDH+ cells in HMLE cells with EGFL9 expression and 250 and 500 nM of JNJ38877605 treatment. **c** ALDH+ FACS profiles are shown for HMLE/LacZ cells, and HMLE/EGFL9 cells with different doses of 2-DG treatment. **d** FACS analysis of the ALDH+ fraction cells in HMLE cells with EGFL9 expression and 5 and 20 mM of 2-DG treatment. **e** Tumorsphere formation assay of HMLE/EGFL9 ($1 \times 10^4$ and $0.5 \times 10^4$) cells with JNJ38877605 and/or 2-DG treatment. **f**, **g** The effect of EGFL9 on drug resistance in HMLE cells. Different doses of Dox (**f**) and Pac (**g**) were used to treat HMLE/LacZ and HMLE/EGFL9 cells. MTT assay was used to measured cell viability. For panels from **e** to **g**, each bar represents the mean ± SEM (standard error of the mean of a representative experiment performed in triplicate). $P$ values were determined by unpaired two-tailed $t$-test. *$P < 0.05$; **$P < 0.01$ and ***$P < 0.001$. **h**, **i** Treatment of JNJ38877605 (500 nM) and 2-DG (5 mM) in combination sensitizes HMLE/EGFL9 cells to Dox (**h**) and Pac (**i**) treatment. **j**, **k** Treatment of JNJ38877605 (500 nM) and 2-DG (5 mM) in combination sensitize SUM159 cells to Dox (**j**) and Pac (**k**) treatment. For panels from **h** to **k**, two-way ANOVA followed by multiple comparisons was used to compare the two values at a given concentration. *$P < 0.05$; **$P < 0.01$; and ***$P < 0.001$

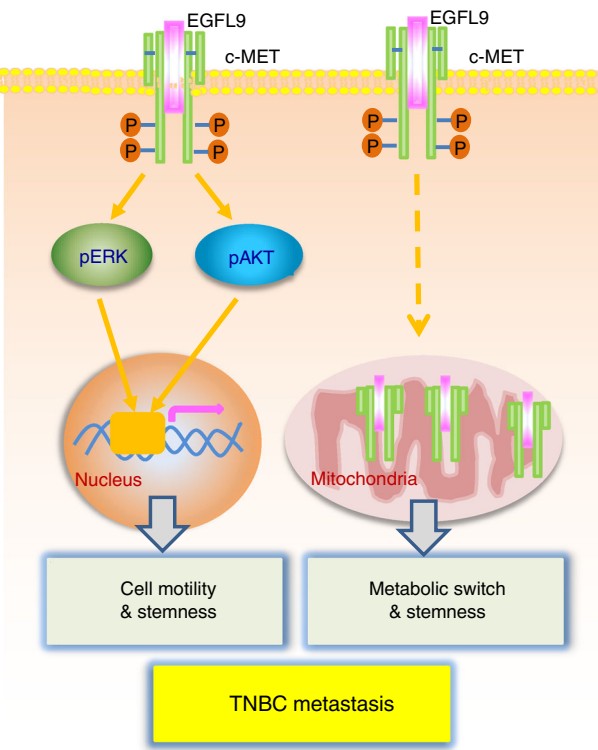

**Fig. 9** Schematic illustration of the EGFL9 in driving TNBC metastasis

upon EGFL9 expression and cMET activation has been reported to contribute to cancer metastasis[47,48]. Second, cell surface molecular interactions have been frequently reported as a mechanism to activate RTKs, including EGFR and cMET[49,50]. In future studies, we will focus on precisely deciphering the structure of EGFL9–cMET interaction and identifying critical fragments or amino acids in EGFL9 protein using a combined genetic and biochemical approach. The mutant EGFL9, which is incapable of interacting with cMET, will then be examined for its effect on cMET activation and distant metastasis.

Another unanswered question in our study is how EGFL9 and cMET translocate to mitochondria. Accumulating evidence show that endocytosis is the major regulator of signaling from receptor tyrosine kinases (RTKs)[51]. The canonical model of RTK endocytosis involves rapid internalization of an RTK activated by ligand binding at the cell surface and subsequent sorting of internalized ligand-RTK complexes to lysosomes for degradation. In addition, translocation of these RTKs to the nucleus and mitochondria has also been reported. Specifically, EGFR has been shown to translocate to mitochondria through both endocytosis dependent and independent mechanisms[45,52]. Her2 protein was reported to translocate into mitochondria by interacting with HSP70[46]. In the future, we will explore if EGFL9 and cMET translocate to mitochondria via an endocytosis-linked mechanism and whether the translocation is chaperone dependent.

Our current study has three potential clinical applications. First, cMET activation has been recognized as a mechanism for drug resistance to anti-EGFR therapy[53–55]. Due to the function of EGFL9 as an activator of cMET, EGFL9 expression may be a predictive marker for resistance to EGFR-targeted therapy in some breast cancer patients. Second, anti-cMET therapy has been developed for different cancers, including invasive breast cancer[47,56–59]. Unfortunately, none of these therapies improved overall patient survival[60–62]. A closer examination of c-Met trials showed that 71% of the cMET trials did not indicate the use of gene or protein markers[63], suggesting that the lack of tumor response could be due to improper selection of patients. As EGFL9 specifically activates cMET, EGFL9 may be a promising marker to stratify breast cancer patients for cMET-targeted therapy. Finally, based on the functional role of EGFL9 and its underlying mechanisms, targeting EGFL9 signaling by blocking the EGFL9/cMET interaction or combining anti-cMET therapy with glycolysis inhibition may offer additional approaches for TNBC treatment.

## Methods

**Cell culture**. All human breast cancer cell lines were obtained from and characterized by cytogenetic analysis by American Type Culture Collection (ATCC, Manassas, VA). All cell lines were grown according to ATCC recommendations. All of the cell lines were authenticated upon receipt by comparing them to the original morphological and growth characteristics and were confirmed to be mycoplasma-free. The mouse mammary epithelial cell line EpRas and the human mammary epithelial cell line HMLE were obtained from Dr. Robert A. Weinberg's laboratory at MIT. EpRas was maintained in DMEM, 8% FBS, and 500 µg/ml G418. HMLE cells were maintained in the culture as previously described[64]. The presence of the H-*Ras* oncogene in the EpRas cell line, as well as the SV40 large T antigen and a catalytic subunit of telomerase in the HMLE cell line, was confirmed by PCR. The mouse breast cancer cell line 4T1 was originally generated at the Karmanos Cancer Institute and has been characterized for its metastatic capability in vivo[65]. The 4T1 cells were cultured in high glucose DMEM supplemented with 5% FBS, 5% NCS, NEAA, and antibiotics (100 U/ml penicillin and 100 µg/ml streptomycin). For studying bioenergetics, HMLE/EGFL9 cells were cultured with low-glucose DMEM mixed with F12 medium (1:1) and supplemented with other reagents as in HMLE cells. SUM159 cells were cultured in F12 medium supplemented with 5% FBS, 5 µg/ml insulin, and 1 µg/ml Hydrocortisone. These culture conditions minimize the effect of glucose on bioenergetics switch. For drug treatment experiments, cells were treated with different doses of JNJ38877605 (cMET inhibitor; Selleck Chemicals), 2-DG/NHI-2 (glycolysis inhibitor; Sigma), or combination of these two drugs. Cell lysates were then collected with RIPA buffer and subjected to western blotting, or the cells were used for MTT assay or Tumorsphere assay.

**Western blotting**. Cells were lysed with cell extraction buffer (Life Technology, Carlsbad, CA) supplemented with 1× protease and phosphatase inhibitor cocktail (Thermo Fisher Scientific, Grand Island, NY). Thirty to 50 µg of total protein from each sample was resolved on a 6–10% Bis-Tris gel with Tris/glycine/SDS running buffer and transferred to nitrocellulose membranes (Bio-Rad Laboratories, Hercules, CA). The blots were then probed with various antibodies. Antibodies for p-cMET (Tyr1234/5, #3077, 1:500), cMET (#3127, 1:1000), p-EGFR (#2237, 1:500), EGFR (#4267, 1:500), p-FAK (Y925, #3284, 1:500), FAK (#3285, 1:1000), pAKT (Ser473, #9271, 1:1000), AKT (#9272, 1:1000#), p-ERK (Thr202/Tyr204, #9101, 1:1000), ERK1/2 (#9102, 1:1000), Sox2 (#2748, 1:500), Nanog (#4903, 1:500), and Oct4 (#2840, 1:500) were all obtained from Cell Signaling Technology (Danvers, MA). Antibody for EGFL9 (A12078, 1:750) was purchased from Booster Biological Technology (Pleasanton, CA). V5 antibody (#46-0705, 1:5000) was purchased from Life Technology. Antibodies for E-cadherin (#610404), α-catenin (#610193), β-catenin (#610153), γ-catenin (#610253), and fibronectin (#610077) were obtained from BD Biosciences (San Jose, CA). Vimentin (sc-6260, 1:1000) and β-actin (sc-47778, 1:2000) were obtained from Santa Cruz Biotechnology (Dallas, TX).

The cytosolic and mitochondrial fractionations used for western blotting were collected using the cell fractionation kit from Abcam (ab109719). Briefly, cells were seeded in 10 cm Petri dishes and were trypsinized when their confluence reached 80–90%. Cells were then washed and followed by a two-step lysis with buffers supplemented with different detergent. Cytosolic and mitochondrial fractions were subsequently collected after each lysis. The protein concentration within each fraction was quantified using Bradford assay. About 20 µg protein from each fraction of each cell line was separated by SDS-PAGE and electro-transferred to nitrocellulose membranes (Bio-Rad Laboratories, Hercules, CA). Phosphorylated c-Met and total c-Met expression was detected by phospho-c-Met antibody (Thr1234/5; Cell Signaling, #3077, 1:500) and c-Met antibody (Cell Signaling, #3127, 1:1000). EGFL9 was detected by an antibody from Booster (A12078, 1:750). Beta-actin (Santa Cruz, sc-47778, 1:2000) was used as the cytosolic control and cytochrome *c* (BD Biosciences, 556433, 1:3000) was used as control for the mitochondrial fraction.

After overnight incubation with the primary antibody at 4 °C, membranes were washed with TBS/T and incubated with anti-mouse IgG (GE Healthcare, NXA931V) or anti-rabbit IgG (Cell Signaling #7074) horseradish peroxidase-conjugated secondary antibody for 1 h at room temperature. Bands were developed by adding horseradish peroxidase substrate (Thermo Fisher) and recorded with X-ray film or ChemiDoc™ digital imaging system from Bio-Rad. Uncropped scans of the most important western blots are shown in Supplementary Fig. 14.

**Human Phospho-RTK array and tissue microarray**. The human Phospho-RTK array kit was purchased from R&D Systems (ARY001B, Minneapolis, MN, USA),

which contains 49 RTK phosphorylation antibodies. Briefly, the array was incubated with 300 µg whole-cell lysates overnight at 4 °C with shaking and washed with the supplied washing buffer. Arrays were then incubated with anti-phosphotyrosine-HRP antibodies for 2 h at room temperature on a rocking platform shaker before incubation with a chemiluminescent reagent. The arrays were scanned with the Bio-Rad Molecular Imager Gel Doc XR system. The final results were quantitated with the Image J software.

A breast cancer tissue array, 150 cores including 5 cases of normal control and 70 cases of reactive, premalignant, and malignant tissues of the breast in duplicates were purchased from Pantomics Inc. (Richmond, CA). Standard procedures were used for the immunohistochemistry (IHC) assay. The EGFL9 antibody was purchased from Abnova (Walnut, CA, #H00065989-B01) and the working dilution was 1:100.

**RT-PCR and quantitative real-time PCR**. RT-PCR was performed as previously described[66]. Briefly, a total of 1 µg of RNA from each cell line was used to generate single-strand cDNA with random hexamer primers using the Superscript II first-strand synthesis system for RT-PCR (Invitrogen, Carlsbad, CA). Quantitative real-time-PCR was done using the iQSYBR Green Supermix (Bio-Rad). A GAPDH primer set was used as an internal control. All the RT-PCR and quantitative real-time PCR primer sequences are provided in Supplementary Tables 2 and 4.

**Generation of expression constructor and stable cell models**. A plasmid containing full-length *EGFL9* was purchased from Open Biosystems (Huntsville, AL). The *EGFL9* gene was then subcloned into a pEntr vector and recombined into a pLenti-6 Vector. A set of five shRNA clones for *EGFL9* was purchased from Open Biosystems. The lentivirus for both overexpression and shRNA targeting *EGFL9* were generated using the pLenti-virus-expression system (Invitrogen, Carlsbad, CA) and the Trans-Lentiviral packaging system (Open Biosystems). It was then used to infect the targeted model cells. Stable cells were generated after being selected with Blasticidin (10 µg/ml) or Puromycin (12 µg/ml) (Invivogen, San Diego, CA) for 20 days. All the primer sequences for cloning of EGFL9 are provided in Supplementary Table 2. All the shRNA sequences are provided in Supplementary Table 3.

**Tumorsphere formation assay**. A mammosphere formation assay was performed as previously described[67] with the following modifications. Briefly, 10,000 cells were plated on a six-well ultra-low attachment plate (Corning Inc., Kennebunk, ME) and were grown in serum-free mammary epithelial growth medium (MEBM Basal Medium, Lonza) supplemented with B27 (Invitrogen), 20 ng/ml EGF, 1 µg/ml hydrocortisone, 5 µg/ml insulin, and 5 µg/ml β-mercaptoethanol. One milliliter of medium was added every other day for seven to 12 days. Images of mammospheres were recorded and the number of mammospheres was manually counted. Experiments were performed in triplicate and repeated two times.

**Cell migration and invasion assays**. The in vitro migration and invasion assays were performed as described previously[68] with minor modifications. Cell migration and invasion assays were performed using the 24-well control chamber or Matrigel invasion chamber respectively according to the manufacturer's instructions (BD Biosciences, San Jose, CA). Cells were seeded in a density of $2.5 \times 10^4$/chamber with regular medium without FBS. Full medium with 10% FBS was used as a chemoattractant. Twelve to 24 h after seeding, migrating or invading cells were fixed and stained with the Diff-Quik kit.

**Cell proliferation assay**. For the cell proliferation assay, cells were seeded in triplicate at the density of $2.5 \times 10^3$ per well in 96-well plates on day 0. Cell proliferation was measured with an MTT assay kit (Fisher Scientific, Pittsburgh, PA) on days 1, 3, 5, 7, and 9. All these experiments were repeated at least three times.

**Tumor growth and metastasis assay**. All animal handling and procedures were approved by Wayne State University Institutional Animal Care and Use Committee (IACUC, 15-12-026). Purchased mice were randomly divided into two groups upon reception. Each group consisted of five mice. Sample size estimate was based on xenograft assays from literatures and our previous studies. Since tumors were observed in all the mice injected with cells, no mice were removed from our study. EpRas-derived tumor cells ($2 \times 10^5$) in 100 µl PBS were injected into the mammary glands of 5-week old, female NCR Nu/Nu mice that were purchased from Taconics (Hudson, NY). 4T1-derived tumor cells ($2 \times 10^4$) in 50 µl were injected into the mammary glands of 5-week old female BALB/C mice that were purchased from The Jackson Laboratory (Bar Harbor, MA). Post tumor cell injections, mice were sacrificed at the fourth week. The mammary tumor and lungs were removed and embedded into paraffin blocks. Standard H&E staining of a series of five sections of paraffin-embedded tissue was performed for histological examination of metastases. The number of microscopically visible pulmonary metastases was obtained by using bright-field microscopy. The measurement and data processing were done by an investigator blinded to cell injection procedure.

**In vivo MRI**. The mice were continuously anesthetized with isoflurane (1.5% of isoflurane/air (v/v)). Their body temperature was maintained at 37 °C during MRI scanning. The mice were imaged in a 7T 60-mm vertical-bore micro-imaging system (clinicSpin, Bruker) using a single-turn radio-frequency (RF) surface coil. For the primary tumor, a coronal fat-suppression T2-weighted imaging sequence (TR/TE = 2200 ms/32 ms, 18 slices, slice thickness = 1 mm with no gap, image matrix size = 256 × 256, field of view (FOV) = 35 × 30 mm, NEX = 2 averages) was performed. For detection of lung metastasis, an axial and a coronal fat-suppression T2-weighted imaging sequence (TR/TE = 3200 ms/26 ms, 18–32 slices, slice thickness = 0.8 mm with no gap, image matrix size = 192 × 192, field of view (FOV) = 36 × 28 mm, NEX = 8 averages) were performed. The total imaging time was 10 min. After MRI scanning, the volume of the primary tumor and the number of lung metastases and lymph node metastases were evaluated independently using an imaging workstation.

**Co-immunoprecipitation**. Cellular lysates were prepared by incubating the cells in lysis buffer (10 mM CHAPS, 50 mM Tris-HCL, pH 8, 150 mM NaCl, 0.5% NP-40, 2 mM EDTA) containing a protease inhibitor cocktail (Thermo Fisher Scientific) for 20 min at 4 °C, followed by centrifugation at 14,000g for 15 min at 4 °C. The protein concentration of the lysates was determined using the Bradford protein assay kit (Bio-Rad) according to the manufacturer's protocol. Ten percent (1:10) cellular extracts were used for input. For immunoprecipitation, 500 µg of protein was incubated with 2 µg specific antibodies for 12 h at 4 °C with constant rotation; 60 µl of 50% protein A or G agarose beads was then added and the incubation was continued for an additional 2 h. Beads were then washed five times using CHAPS co-IP lysis buffer. Between washes, the beads were collected by centrifugation at 500g for 5 min at 4 °C. The precipitated proteins were eluted from the beads by resuspending the beads in 2× SDS-PAGE loading buffer and boiling for 10 min. The resultant materials from immunoprecipitation or cell lysates were resolved using 10% SDS-PAGE gels and transferred onto nitrocellulose membranes for western blotting. To examine EGFL9 and cMET interaction, reciprocal co-IP was performed in HMLE/EGFL9 cells with anti-cMET (Cell Signaling, #3127) or V5 (Life Technology, #46-0705) antibody, respectively. Western blotting was then performed with anti-V5 or cMET antibody accordingly. In SUM159 cells, co-IP was performed with anti-cMET antibody and followed with western blotting with anti-EGFL9 antibody. To detect COX2 phosphorylation levels, HMLE/LACZ and HMLE/EGFL9 were serum starved for 24 h prior to harvest in CHAPS IP buffer. Lysates (500 µg) were subject to pull-down with 2 µg COX IV antibody (Santa Cruz, sc-69360) and samples were run on 15% gel; membranes were probed for COX II (ProteinTech, #55070-1-AP) and antibody against pan-phosphorylated tyrosine (Millipore, #05-321). To confirm EGFL9 and COA3 interaction, pLent-6-EGFL9-V5 and p3xFlag-CMV-COA3 constructs were co-transfected in 293T cells. Co-IP was performed with anti-V5 (Life Technology, #46-0705) or antibody. Western blotting was then performed with anti-Flag (SIGMA, #F7425).

**Immunofluorescence and confocal microscopy**. Cells growing on cover slips in a six-well plate were washed with PBS and fixed in 4% (w/v) paraformaldehyde, permeabilized with 0.1% (v/v) Triton X-100 in PBS, blocked with 0.8% BSA, and incubated with appropriate primary antibodies followed by staining with Alexa Fluor 488 (#A11001, 1:500) or Alexa Fluor 594 (#A11012, 1:500)-conjugated secondary antibodies (Molecular Probes, Invitrogen). The cells were washed four times, immersed in a drop of ProLong Gold antifade reagent with DAPI (Thermo Fisher Scientific, #P36941), and sealed on slides. To check the expression of E-cadherin (BD, #610181), Vimentin (Santa Cruz, #sc-6260), phosphor-cMET (Cell Signaling, #3077), phosphor-ERK (Cell Signaling, #9101), HMLE/LacZ, and HMLE/EGFL9 or SUM159/NT, SUM159 sh2 and SUM159 sh3 cells were used. Images were visualized with an Olympus Eclipse Ti-E inverted microscope equipped with a charge-coupled camera.

For study the co-localization of EGFL9 and cMET in mitochondria, 100 µM MitoTracker™ Deep Red FM dye (Thermo Fisher Scientific, #M22426) was used in HMLE/EGFL9 or SUM159 cells for 45 min to stain mitochondria. Then the cells were stained for EGFL9-V5 (Life Technology, #46-0705), cMET (Cell Signaling, #3127) following the regular immunofluorescence protocol as described above.

For quantification of protein co-localization, confocal z-stacks of immune-labeled sections were imported into Volocity (6.3.1) high-performance 3D imaging software. Individual cells were analyzed using the freehand ROI tool and cropped to selection. Regions of the image that contained no visual signal (background) were selected and used for thresholding the image. Co-localization analysis was performed using the co-localization function within Volocity producing co-localization coefficients.

**Biomolecular fluorescence complementation assay**. Bimolecular fluorescence complementation (BiFC) constructs were made by fusing VN (Venus N1-173) to the C-terminus of EGFL9 (NM_023932.3). A linker, GGSGSGSS, was inserted between the Venus tag and EGFL9. The fusion molecule was then subcloned into p3XFLAG-CMV vector. Similarly, VC (Venus C155-239) was fused to the C-terminus of COA3 (BC002698) with a linker, which was then subcloned into p3XFLAG-CMV vector. For BiFC assay, 293T cells were transfected with 1 µg of each plasmid (p3xFLAG-CMV-EGFL9-VN and p3xFLAG-CMV-COA3-VC) using

Lipofectamine 2000 for 48 h. Live-cell imaging was performed using a Zeiss LSM 510 confocal microscope. We used a Plan-Apochromat ×63/1.4 oil objective using both a zoom of 1 and 1.6. The blue channel was imaged using a 745 nm Coherent Chamelean Ti:Sa multiphoton laser for excitation and a 390–465 nm band pass filter for emission gating. The green channel was imaged using a 488 nm Argon laser for excitation and 500–550 nm band pass filter for emission gating. The red channel was imaged using a 545 nm HeNe laser for excitation and 565–615 nm band pass filter for emission gating. The purple channel was imaged using a 633 nm HeNe laser for excitation and 650–710 nm band pass filter for emission gating. For the movies we used Volocity Analysis Software (Version 6.3.1, Perkin Elmer).

**Isolation of mitochondria and proteinase K protection assay**. The mitochondria from about $3.5 \times 10^7$ of HMLE/EGFL9 cells were isolated using the Mitochondria Isolation Kit (Thermo 89874). The isolated mitochondria were re-suspended in buffer supplemented with 0.25 M sucrose, 20 mM Tris·HCl, and 0.15 mM $MgCl_2$ and split into five equal fractions. Low (0.5 μg/ml) or high (2.0 μg/ml) concentration of proteinase K with or without digitonin (5 mg/ml) was applied to each fraction of isolated mitochondria for 30 min in ice water. The treatment was terminated by adding 1 mM PMSF and incubation for 5 min at 4 °C. Then equal amount of cell lysis buffer (FNN0011; Life Technology) with protease inhibitor cocktail were added and incubated on ice for 30 min. SDS loading buffer with beta-mercaptoethanol was added and heated at 95 °C for 10 min. One-third of the lysis fraction was loaded on a 12% SDS-PAGE gel. Western blotting was performed for V5 (Life Technology, #46-0705, 1:4000), cMET (Cell Signaling, #3127, 1:500), Tom20 (Cell Signaling, #72610, 1:1500), HSP60 (Proteintech, #15282-1-AP, 1:2000), and Cytochrome C (BD, #556433, 1:2500).

**COX-specific activity assay**. Mitochondria were isolated from HMLE/LACZ and /EGFL9 cell lines using the Mitochondrial Isolation Kit for Cultured Cells (Thermo Fisher Scientifics). The Cytochrome Oxidase Activity Colorimetric Assay Kit from Biovision was used to measure cytochrome oxidase activity. In all, 2.5 μg of mitochondria were plated in a 96-well plate with a mixture of reduced cytochrome oxidase and buffer per manufacturer instructions. Then, the OD at 550 nm was measured and the maximum linear decay was used to calculate activity (units/mg). The difference between groups was evaluated for statistical significance by a two-sample $T$-test. The assay was performed in triplicate, with two blank wells containing only the reduced cytochrome c/buffer mixture as a negative control.

**OCR measurements**. OCR was measured using an XF24 extracellular flux analyzer (Seahorse Biosciences). Cells were seeded in an XF24-well cell culture plate (Seahorse Bioscience) at a density of $20 \times 10^3$ cells/well in 300 μl of DMEM-Glu medium and incubated for 24 h at 37 °C, 5% $CO_2$. The following day the medium was replaced with fresh DMEM-Glu. After 48 h, the growth medium was changed to 500 μl of unbuffered DMEM, pH 7.4, and cells were incubated at 37 °C without $CO_2$ for 1 h for equilibration before starting the assay. OCR was measured for ~3 min in repeated cycles, to obtain basal rates. After baseline measurements, OCR was measured after the injection of Oligomycin to a final concentration of 1 μM. Subsequently, carbonyl cyanide-4-(trifluoromethoxy)phenylhydrazone (FCCP) was injected to reach a working concentration of 5 μM and maximal OCR was measured. Non-mitochondrial oxygen consumption was measured after the injection of Rotenone and Antimycin A to a final concentration of 3 μM each. All OCR values were normalized to total protein as determined with the DC protein assay kit (Bio-Rad).

**Lactate production analysis**. Lactate level was analyzed with a glycolysis cell-based assay kit (Cayman Chemical). Briefly, 10,000 of HMLE/LacZ or HMLE/EGL9 were seeded in 96 wells. The next day, the plate was centrifuged for 5 min and 10 μl of the supernatant from each well was taken and added to the assay plate and incubated with glycolysis substrate and cofactor for 30 min at room temperature. The absorbance at 490 nm was measured and the lactate generated was quantified per the standard curve.

**Glucose uptake analysis**. The glucose concentration was measured with the Amplex Red Glucose Assay Kit (Life technology MP22189). Briefly, about $2 \times 10^5$ of each cell line was seeded into a well of a 24-well plate. The medium from each well was collected at given time points and was diluted 200 times. A volume of 50 μl of the diluted medium from each treatment was added to a 96-well plate. Another 50 μl of working solution containing 100 μM Amplex Red reagent, 0.2 U/ml HRP, and 2 U/ml glucose oxidase was added to each wells. The plate was incubated at room temperature for 30 min and the absorbance at 560 nm was recorded on a plate reader (SpectraMax i3x from Molecular Devices). The glucose concentration from each sample was calculated according to the standard curve. The difference of the glucose concentration between the initial medium and each sample is the glucose uptakes of each experimental sample.

**Mitochondrial membrane potential analysis**. To measure the mitochondrial potential, $2 \times 10^5$ of HMLE/LacZ or HMLE/EGFL9 cells were seeded into a six-well plate the day before measurement. The next day, cells including a well of HMLE/

EGFL9 cells pretreated with CCCP (50 μM) for 30 min were trypsinized, washed, and re-suspended into prewarmed medium with 10 μg/ml JC-1 and incubated at 37 °C for 30 min. The cells were then washed and analyzed with a BD LSR II flow cytometer to read the signal at Ex 475 ± 20 nm Em 530 ± 15 nm, and 590 ± 17.5 nm.

**ROS analysis**. In total, 10,000 of HMLE/LacZ or HMLE/EGL9 cells were seed in 96 wells. Twenty-four hours later, the cells were washed with PBS and incubated with PBS containing DCFDA (10 μM, for whole-cell ROS) or Mitosox (5 μM, for mitochondrial superoxide) for 45 or 10 min, respectively. Cells were then washed and measured with a fluorescence plate reader under 485/535 nm for whole-cell ROS or 510/580 nm for mitochondrial superoxide.

**ATP production assay**. ATP generation was measured using an ATP detection assay kit (Cayman Chemical). Briefly, about 100,000 of HMLE/LacZ or HMLE/EGFL9 cells were seeded into the six-well plate the day before the test. The cells were then rinsed with PBS and lysed with ATP sample buffer and 10 μl of each cell lysate were added to the 96-well plate. Another 100 μl of reaction mixture containing D-Luciferin and luciferase was added and incubate 20 min at room temperature. The luminescence was measured by a SpectraMax i3x from Molecular Devices. The amount of the ATP generated by the cells was calculated per the standard curve and was normalized based on the protein concentrations.

To assess contribution of mitochondrial OXPHOS and glycolysis to ATP production, about 4000 of HMLE/LacZ or HMLE/EGFL9 cells were seeded in a 96-well plate. The next day, cells were treated with two inhibitors of mitochondria respiratory chain complex Oligomycin (10 μM) or Rotenone (10 μM) for 30 min. The intracellular ATP level was measured following the protocol of ATP detection assay kit as above.

**Flow cytometry analysis**. The ALDEFLUOR kit (StemCell Technologies, Durham, NC, USA) was used to isolate the cell subpopulation with ALDH enzymatic activity. Fresh cells were suspended in ALDEFLUOR assay buffer containing ALDH substrate (BAAA, 1 μM/L per $1 \times 10^6$ cells) and incubated during 40 min at 37 °C. An aliquot of each sample of cells was treated with 50 mM/L diethylaminobenzaldehyde (DEAB), a specific ALDH inhibitor, to establish the baseline fluorescence. All the cell samples were analyzed by BD LSR II flow cytometry.

**Statistical analysis**. A two-sided independent $t$-test without equal variance assumption was performed to analyze the results of cell growth, colony formation, cell migration and invasion, tumor burdens and tumor metastasis results. Two-sided Cochran-Armitage trend test was used to assess linear trend in proportions of binary conditions across the ordinal level of EGFL9 IHC intensities from TMA analysis. The proportions were estimated with Wilson's 95% CI. For the dose-outcome plots for Fig. 8h through k: two-way ANOVA followed by multiple comparisons was used to compare the two values at a given concentration. $P$ value less than 0.05 is considered significant. R version 3.4.3 was used.

**Reporting summary**. Further information on research design is available in the Nature Research Reporting Summary linked to this article.

## Data availability

The source data underlying Figs. 1a, 2a–h, 3a–c, 4c, e, f, 5e, 7c, d, f–i, 8e, f, h–k, and Supplementary Figs. 1, 2e, g, 8d, e, 10c, d, e, 11c, d, 12c, 12d, h, 13a are provided as a Source Data file. Unprocessed original scans of blots are shown in Supplementary Fig. 14. The remaining data are contained within the article, Supplementary Information or available from the authors upon request. A reporting summary for this article is available as a Supplementary Information file.

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

## Acknowledgements

We want to thank Dr. Robert A. Weinberg at MIT for providing the HMLE and EpRas cell lines and Drs. Larry H. Matherly and Manohar Ratnam in the Karmanos Cancer Institute for advice. We also want to thank Ms. Elizabeth A. Katz of the Karmanos Cancer Institute's Marketing & Communications Department for editing our manuscript. This work was supported in part by the NIH RO1 grant CA172480-01A1 (to G.W.); MT program pilot grant (25RU21), and cancer metabolism discovery grant (25SYT) from the Karmanos Cancer Institute, Wayne State University (to G.W.); the National Natural Science Foundation of China (Grant No. 81572601, to F.M. and Grant No. 81272377 to H.Z.); the Natural Science Foundation of Jiangsu Province (Grant No. BK20151095) (to F.M.), NIH training grant T32CA9009531 (to A.M. and C.J.B.); and NCI individual predoctoral fellowship F31CA236245 (to A.M.). The Biostatistics Core, the Microscopy, Imaging and Cytometry Resources Core and the Biorepository Core of the Karmanos Cancer Institute are supported by NIH Center grant P30-CA022453-29. The Microscopy, Imaging and Cytometry Resources Core is also supported, in part, by the Perinatology Research Branch of the National Institutes of Child Health and Development at Wayne State University.

## Author contributions

F.M., L.W., L.D., J.W., M.H. and G.W. conceived and designed all experiments. A.V.M. conducted co-immunoprecipitation analysis and western blotting. F.M. and L.W. performed the in vivo tumorigenesis and metastasis experiments and IHC analysis. J.L. and M.H. performed COX activity analysis and the OCR assay. Q.L. and J.H. performed MRI experiments. W.C., W.-m.S., and B.Z. performed all of the statistical analyses and bioinformatics analyses in this study. F.M., L.W., L.D. and H.Z. performed RT-PCR, western blot, immunofluorescence analysis, and all the other experiments. F.M., L.W., L.D., A.V.M., C.J.B., M.H., Q.Y., J.W. and G.W. discussed the data. G.W. supervised and wrote the final draft of the manuscript.

## Competing interests

The authors declare no competing interests.
