## [Peer Review File · Nature Communications]

Reviewers' comments:

Reviewer #1 : Mitochondria biology
(Remarks to the Author):

This MS by Guojun Wu and Maik Huttemann groups attempt to show that physical interaction between EGFL9 and cMET interaction in metastasis and chemoresistance of human TNBC breast cancers. Using a series of breast cancer cell lines with varying levels of metastatic potential and shRNA mediated knock down of mRNA, they show some level of correlation between EGFL9 expression and metastasis. They then attempt to show direct interaction of EGL9 with cMET in the perinuclear region of cells and possibly their migration to the mitochondrial membrane compartment affects mitochondrial respiration and induces lactate production. The latter part of the work is not well carried out. Despite addressing an interesting research topic, additional data would be required to substantiate their conclusion.

1. The authors claim EGFL9 expression correlates with chemoresistance. They need to support this with TCGA or other cancer genome dataset analysis. It would be interesting to show correlation between EGFL9 expression and chemoresistance to Platinum based drugs, and or combinations with PARP-1 inhibitors.
2. How does EGFL9 correlate with BRCA1, which accounts for 15-20% of the basal cell TNBCS?
3. Most of the experiments are done using transformed cell lines (cancer and non-cancer) which commonly undergo genetic drift in cell culture conditions and could contribute to the findings related to the EMT and metastasis outcome. Therefore, to convincingly establish the role of EGFL9 in metastasis, the authors need to do the metastasis assays using basal-cells derived from patient biopsies cultured ex vivo in xenograft studies in mouse models.
4. Authors have not demonstrated convincingly the role of EGFL9 in mitochondria/metabolic reprogramming. Authors should use stringent imaging techniques such as tag EGFL9 and the mitochondrial gene with photoactivable vectors to show colocalization. In the absence of that, it could be just technical artifacts due to imaging depth or antibody issues. Z-stack images and more details on the imaging parameters should be provided.
5. If the authors imply intramitochondrial localization of EGFL9 and cMET, at the minimum a protease protection analysis would be required. Digitonin fractionation is another useful approach.
6. With the current data provided, the effect of EGFL9 in mitochondrial functions (respiration) and subsequent functions can be attributed to secondary / indirect effects of this factor on cellular functions in general. Moreover the moderate 1.5 fold increase in glucose uptake or lactate production shown in fig7 is only marginal and cannot support the authors' claims about metabolic reprogramming.
7. The Seahorse respirometry in Fig. 7 is poorly done. Why is there a major difference in basal respiration between EGFL9 expression and LacZ expression cells? Here it is imperative to measure ADP dependent respiration and also ECAR (extracellular acidification) which is the direct reflection of metabolic switch to glycolysis.
8. It would be useful to know the ATP production in fully energized cells and cells with complete membrane depolarization to understand the extent of contribution by glycolysis.
9. In Fig. 1C, it is not clear if there is a correlation between mRNA level and protein level. Based on the immunoblot in Fig. 1C, one can not say that EGFL9 levels correlate with the metastatic potential.
10. In Figure 5 and elsewhere, the immunoblots lack appropriate loading controls.
11. The physical association between EGFL9 and cMET should be further supported by mutational analysis.

Reviewer #2: Cancer metabolism and metastasis
(Remarks to the Author):

This manuscript elucidates the role of EGFL9 in basal-like TNBCs. The authors show that EGFL9 is over-expressed in basal-like breast cancer cells and correlates with the metastatic potential of cell lines. They demonstrate that EGFL9 promotes migration and invasion in vitro and metastasis in vivo. Overexpression of EGFL9 induced phosphorylation of cMet and activation of cMet pathway through direct interaction. EGFL9 overexpression also increases ROS and lactate production, while decreasing COX activity and the oxygen consumption rate of the tumor cells expressing EGFL9. They conclude that EGFL9 induces a metabolic switch from oxidative phosphorylation to glycolysis. This paper uses strong and convincing data to support their conclusions. I have a few minor suggestions for the improvement of this paper before it gets accepted. In figure 3B and 4D, the authors should include the number of metastatic nodules per lung beside the percentage the metastasis/lung.

In figure 4C and D, the data will be better represented, if the authors could provide a bar graph instead of a table to show the difference in tumor size with error bars and significance values.

In figure 5D, the changes in pAKT, pERK by western blot are not convincing and it is not adding any value to the conclusion. Either the authors could remove these two blots or repeat.

In figure 6B, the authors don't see a clear expression of endogenous cMet or EGFL9, which makes sense because 10% input may not be enough to see the endogenous proteins, but they do see a striking interaction using this pull down. However, one would expect for sure to see the overexpressed proteins in 10% input, but it is not the case. One possibility that the panels are switched. The authors need to reconfirm these results by repeating it.

In figure 8K, the authors should be consistent in labeling single agent treatment in blue or the same color across.

Reviewer #3: Breast Cancer metastasis
(Remarks to the Author):

In the current study Meng et. al. identified that EGFL9 promotes breast cancer metastasis by inducing cMET activation and metabolic reprogramming. They also found that EGFL9 enhances stemness and chemoresistance in breast cancer. Moreover, targeting cMET signaling and metabolic reprogramming synergistically sensitizes breast cancer cells to chemotherapeutic agents. The findings described in this study are novel, interesting and clinically relevant. However, there are several minor points need to be addressed or discussed before publication.

1. In Fig.1, the mRNA levels of EGFL9 do not correlate well with protein levels across different cell lines. For example, MCF7 has higher mRNA than BT474 (Fig.1A), but lower protein expression (Fig.1B). It would be interesting to investigate the regulation (upstream) of EGFL9. Also, the authors should provide quantitative data for Fig.1B and C.
2. In Fig.1A, EGFL2 is highly expressed in luminal-like cells, which is opposite to EGFL9. It would be interesting to investigate or discuss to see if EGFL2 has opposite biological functions.
3. In Fig.S3A as well as the WB data in the following figures, it would be better to include EGFL9.
4. In Fig. 4C, why Sh3 has significant lower p-cMET and downstream signaling than Sh1? However, these two targets have similar EGFL9 knockdown efficacy (Fig. S2).
5. In Fig. 4D and E, given that p-AKT and p-ERK are not significantly inhibited by the cMET inhibitor JNJ, the treatment time points or concentrations of JNJ need to be further optimized. Also, EGFL9+JNJ treatment group has significant less migrated and invasive cells than LacZ+JNJ group, suggesting EGFL9 may have metastatic suppressive, rather than promoting roles in cells without cMET activation.
6. To confirm EGFL9 promotes metastasis specifically through cMET, downstream signaling should be examined in EGFL9 expressed/KD cells combined with JNJ treatment.

7. Total cMET cannot be observed in the input fractions in Fig. 6A and B. More input should be loaded.

8. Previous study (Dai et. al., Blood 2015 126:2821-2831) suggested that cMET signaling is critical for cell cycle progression. Moreover, this study also suggested cMET is important for cell stemness. It would be better to discuss why EGFL9, which targets cMET, only inhibits metastasis but not primary tumor growth.

Responses to reviewers' comments:

Reviewer #1: Mitochondria biology (Remarks to the Author):

This MS by Guojun Wu and Maik Huttemann groups attempts to show that physical interaction between EGFL9 and cMET interaction in metastasis and chemoresistance of human TNBC breast cancers. Using a series of breast cancer cell lines with varying levels of metastatic potential and shRNA mediated knock down of mRNA, they show some level of correlation between EGFL9 expression and metastasis. They then attempt to show direct interaction of EGL9 with cMET in the perinuclear region of cells and possibly their migration to the mitochondrial membrane compartment affects mitochondrial respiration and induces lactate production. The latter part of the work is not well carried out. Despite addressing an interesting research topic, additional data would be required to substantiate their conclusion.

1. The authors claim EGFL9 expression correlates with chemoresistance. They need to support this with TGCA or other cancer genome dataset analysis. It would be interesting to show correlation between EGFL9 expression and chemoresistance to Platinum based drugs, and or combinations with PARP-1 inhibitors.

Our response: We agree that data mining from existing public resources would provide more insight into the correlative evidence linking aberrance of EGFL9 (DLK2) expression with development of chemoresistance in breast cancer. We thoroughly searched TCGA database and commercial database Oncomine. In order to demonstrate that upregulated EGFL9 leads to worse outcome among patients/samples treated with chemotherapy, we need mRNA data, treatment information, and the outcome. Unfortunately, no available data set provides this information. As an evidence of EGFL9 function in stemness, we observed some effect of combined treatment of JNJ and 2-DG sensitizing EGFL9 overexpression cells to Dox and Pac. In this revision, we removed the word “chemoresistance” and revised our claim to a narrower point of view that is limited to what we observed in our lab only. We hope our paper will serve as a springboard for those who are interested in our hypothesis.

2. How does EGFL9 correlate with BRCA1, which accounts for 15-20% of the basal cell TNBCS?

Our response: This is a very good point. We evaluated the correlation between DLK2 (EGFL9) and BRCA1 in TNBC or basal like breast cancers, whichever was available in TCGA and Oncomine. The results are inconclusive. Briefly, for correlation between DLK2 mRNA expression and BRCA1 mRNA expression:

DataSet	Subtype	Sample Size	Cor. Coef	p. value
TCGA BRCA1 NM_007295_1_6483	TNBC	46	-0.17	0.256
Farmer BRCA1 204531_s_at	Basal -Like	16	-0.3	0.251
Tabchy BRCA1 211851_x_at	TNBC	57	0.57	<0.001
Neve Cell Line BRCA1 204531_s_at	TNBC	21	0.31	0.169
Bittner BRCA1 204531_s_at	TNBC	45	0.09	0.546
Bittner BRCA1 211851_x_at	TNBC	45	0.37	0.013
Zhao BRCA1 IMAGE 2266185	TNBC	5	-0.09	0.891

For evaluating the correlation between DLK2 mRNA expression and BRCA1 mutation in TNBC or Basal-like subgroup, TCGA Provisional has 994 samples available with BRCA1 mutation status, DLK2 expression (RSEM batch normalized from Illumina HiSeq_RNASeqV2), and breast cancer subtypes. Note that the BRCA1 mutation rate in this dataset is a very low 5.5% in TNBC. We found no correlation between EGFL9 and BRCA1 mutation in TCGA.

The above datamining analysis may suggest that BRCA1 and EGFL9 play distinct functions in basal-like breast cancer. Since BRCA1 expression or mutation rate is very low in basal like breast cancer, more patient samples are required to clarify this issue. Since this question is beyond the focus of our current study, we will not include this in this manuscript.

However, by a datamining study in Oncomine, we did find that EGFL9 showed preferential expression in both TNBC cells and tumor samples than in non-TNBC cells and non-TNBC tumor samples in multiple datasets^{1, 2, 3, 4}. These results are consistent with our discovery and were included as new Supplementary Figure S2 in this revision.

3. Most of the experiments are done using transformed cell lines (cancer and non-cancer) which commonly undergo genetic drift in cell culture conditions and could contribute to the findings related to the EMT and metastasis outcome. Therefore, to convincingly establish the role of

EGFL9 in metastasis, the authors need to do the metastasis assays using basal-cells derived from patient biopsies cultured ex vivo in xenograft studies in mouse models.

Our response: We thank the reviewer's advice for utilization of PDX model in our studies. We agree that PDX mouse model is a powerful tool to model certain aspects of tumor growth and progression in addition to the current standard tumor xenograft mouse models⁵, although recent research results demonstrated some shortcomings of PDX model in studying tumor metastasis^{6,7,8}.

However, as the Editor suggested that "Despite recognizing that it would increase the impact of your findings, we would not consider essential that you confirm your results in a PDX model as suggested by Reviewer#1". We hope the reviewer will also agree with editor and allow us to further confirm our findings using PDX model in our future studies. Our current study used two different models to study EGFL9 function in driving metastasis. According to the Editor's suggestion, we discussed the limitation, as well as advantages, of the cell lines used in our current study in the revised text (please see page 16-17, line 432-439).

4. Authors have not demonstrated convincingly the role of EGFL9 in mitochondria/metabolic reprogramming. Authors should use stringent imaging techniques such as tag EGFL9 and the mitochondrial gene with photoactivable vectors to show co-localization. In the absence of that, it could be just technical artifacts due to imaging depth or antibody issues. Z-stack images and more details on the imaging parameters should be provided.

Our response: We thank the reviewer for this constructive comment. To address the concern of reviewer, we purified the mitochondria from HMLE/EGFL9-V5 cells and performed EGFL9-V5 IP mass spectrometry. We identified 23 mitochondrial proteins that were significantly enriched (data not shown). Of interest, we found COA3, a COX assembly factor, as a potential interacting partner of EGFL9 within the mitochondria. We next performed a Biomolecular Fluorescence Complementation (BiFC) assay^{9, 10, 11}. EGFL9 was fused with the Venus N fragment and this fusion molecule was cloned into the p3xFlag-CMV vector. COA3 was fused with Venus-C fragment and was then cloned into the p3xFlag-CMV-vector. These two constructs were then co-transfected into 293T cells. We observed that only cells transfected with EGFL9-VN and COA3-

VC showed apparent VFP signals. COA3 is a COX assembly factor that is localized in the mitochondrial inner membrane. COA3 stabilizes cytochrome *c* oxidase 1 (COX1) and promotes cytochrome *c* oxidase assembly in human mitochondria^{12, 13, 14}. Therefore, our BiFC results provide strong evidence that further illustrates the mitochondrial localization of EGFL9. These results are presented in the new Figure 7a and Supplementary Fig. S9 and Supplementary Movie S3 in our revision.

5. If the authors imply intra-mitochondrial localization of EGFL9 and cMET, at the minimum a protease protection analysis would be required. Digitonin fractionation is another useful approach.

Our response: To comply with the reviewer's request, we have performed a proteinase K protection assay. Different doses (0.5 and 2 μ M) of proteinase K with or without digitonin were applied to equal amounts of purified mitochondria. Lysed proteins were separated by SDS-PAGE and immunoblotted using the antibodies including EGFL9, cMET, Tom20 (outer membrane protein), Cytochrome *c* (an intermembrane space protein) and HSP60 (a matrix protein). Our results demonstrate that HSP60 is fully protected from all the treatments. Tom20 is not protected from proteinase K treatment. While Cytochrome *c* is partially protected from proteinase K treatment, it is digested by combined treatments of Proteinase K and digitonin. EGFL9 and cMET showed a similar pattern with Cytochrome *c*, suggesting both EGFL9 and cMET are localized within mitochondria, in association with the inner membrane. These results are summarized in new Figure 6g.

6. With the current data provided, the effect of EGFL9 in mitochondrial functions (respiration) and subsequent functions can be attributed to secondary / indirect effects of this factor on cellular functions in general. Moreover the moderate 1.5 fold increase in glucose uptake or lactate production shown in fig7 is only marginal and cannot support the authors' claims about metabolic reprogramming

Our response: We understand the reviewer's concern because of the uncertainty of EGFL9 function in mitochondria. However, we believe that several lines of new data provide strong

evidence supporting a direct role of EGFL9 in mitochondrial function (respiration). 1) By using a BiFC assay, EGFL9 was shown to interact with COA3, a critical factor for COX complex assembly (new Fig.7a, Supplementary Fig. S9 and supplementary Movie S3). 2) By using a protease protection analysis combined with digitonin treatment, EGFL9 and cMET localization in mitochondria was further confirmed (new supplementary Fig. 6g). 3) By performing an ATP production measurement in cells treated with two mitochondrial inhibitors, we found that a great portion of ATP production is attributed to glycolysis in cells with EGFL9 overexpression (new Supplementary Fig. S10d).

Our results showed 50% increase in glucose uptake and L-lactate production in EGFL9 overexpression cells as compared to control cells. This data is similar to a recent paper published in *Nature Communication* showing that HER2 expression leads to similar 50% change¹⁵ in glucose uptake and L-lactate production. Taken together we believe the metabolic regulation of EGFL9 on intermediate metabolism as indicated by the alteration of glucose consumption/lactate production is of functional importance.

7. The Seahorse respirometry in Fig. 7 is poorly done. Why is there a major difference in basal respiration between EGFL9 expression and LacZ expression cells? Here it is imperative to measure ADP dependent respiration and also ECAR (extracellular acidification), which is the direct reflection of metabolic switch to glycolysis.

Our response: We are sorry for the confusion. We performed the respirometry analysis using the standard procedure on a XF24 Seahorse flux analyzer. The Seahorse respirometry was performed using stable cell models expressing EGFL9 or LacZ control vector. Since the EGFL9 is constitutively expressed, we expected to see the inhibition of basal respiration in HMLE/EGFL9 cells. Indeed, we observed that elevated EGFL9 significantly suppresses cellular respiratory capacities at both the basal and maximal (shown as the FCCP-induced respiratory phase on the histogram) levels (new Figure 7d). These alterations in cellular respiration are perfectly consistent with our finding that EGFL9 downregulates respiratory complex IV activity (new Figure 7c), likely through EGFL9 binding to the complex IV assembly factor COA3, as currently presented in new Figure 7a and b.

Our respirometry analysis has also revealed that elevated EGFL9 significantly suppresses the cellular respiration in the ADP-dependent phase (shown as the oligomycin-sensitive phase respiration on the histogram) (new Figure 7d), indicating that EGFL9 significantly suppresses the ATP biosynthesis of mitochondrial OXPHOS pathway. In addition, our metabolite analysis clearly demonstrated that elevated EGFL9 significantly increased cellular production and excretion of lactate into culture media. As lactate synthesis and excretion are the unique metabolic steps of aerobic glycolysis and are true metabolic basis for the extracellular acidification that is estimated by measuring ECAR in the Seahorse flux analysis, we believe that these results unequivocally demonstrate that EGFL9 overexpression is associated with suppression of OXPHOS and promotion for aerobic glycolysis in breast cancer cells. Of note, our results are similar to a recent publication in *Nature Communications*, which described the translocation of Her2 oncogene into mitochondria regulating cellular metabolism. In that paper, switching to glycolysis was illustrated by L-lactate production and glucose consumption, without showing any results of ECAR ¹⁵.

8. It would be useful to know the ATP production in fully energized cells and cells with complete membrane depolarization to understand the extent of contribution by glycolysis.

Our response: We thank the reviewer for the constructive advice. Accordingly, we measured the cellular ATP levels in the presence of Rotenone or Oligomycin, which inhibits mitochondrial electron transport chain or ATP synthase, respectively. We found that cells with EGFL9-overexpression express significantly higher levels of ATP compared to the LacZ control cells in the presence of either mitochondrial inhibitor. These results indicate that EGFL9 overexpression significantly boost the production of cellular ATP from the non-mitochondrial source – glycolysis. We present these results in the new Supplementary Fig. S10d.

9. In Fig. 1C, it is not clear if there is a correlation between mRNA level and protein level. Based on the immunoblot in Fig. 1C, one cannot say that EGFL9 levels correlate with the metastatic potential.

Our response: We thank the reviewer pointing this out. We repeated the RT-PCR and Western blot analysis for Figure 1c and got the same results as before. EGFL9 expression was found gradually increase along with the increase in metastasis capability at the RNA level. However, at the protein level, EGFL9 was significantly increased in 168FARN cells and remained at a high level in the other metastatic cell models including 4T07, 4T1 and 66C14. These results may suggest a posttranslational modification of EGFL9 in 168FARN cells. Although we don't think this result contradicts with our claim of EGFL9's correlation with the metastatic potential, we agree to remove the study of mouse cancer cell lines and focused on human cancer cell lines in order not to confuse the readers.

10. In Figure 5 and elsewhere, the immunoblots lack appropriate loading controls.

Our response: We have added all the appropriate loading controls (β -actin) for the immunoblots in Figure 5 and elsewhere. Please see the revised Figure 5.

11. The physical association between EGFL9 and cMET should be further supported by mutational analysis.

Our response: We thank the reviewer for this suggestion. In order to narrow down the protein domains mediating the EGFL9 and cMET interaction, we generated domain-deletion constructs within regions we hypothesized that are critically important for this protein-protein interaction.

For EGFL9 protein, we deleted the EGF-rich domain. By Western blotting, we could not detect expression of EGFL9 mutant protein, suggesting that the EGF-rich domain is essential for EGFL9 protein stability. For the cMET protein, we made a deletion of the SEMA domain (AA 27-515), which has been shown to be responsible for receptor dimerization¹⁶ (see diagram in new Supplementary Fig. S7a). We showed that this deletion did not change the localization of cMET protein (new Supplementary Fig. S7b). When we performed an EGFL9 IP in the presence of either wild-type cMET or cMET with SEMA deletion, we found that deletion of SEMA domain did not alter EGFL9 binding capability (new Supplementary Fig. S7c), suggesting cMET receptor dimerization is not required for EGFL9 binding. While these domain mapping studies

have provided some important insights into the EGFL9-cMET complex, the detailed biochemical mapping of this protein interaction and mutational analysis is something we hope to accomplish in our future work.

**Reviewer #2: Cancer metabolism and metastasis
(Remarks to the Author):**

This manuscript elucidates the role of EGFL9 in basal-like TNBCs. The authors show that EGFL9 is over-expressed in basal-like breast cancer cells and correlates with the metastatic potential of cell lines. They demonstrate that EGFL9 promotes migration and invasion in vitro and metastasis in vivo. Overexpression of EGFL9 induced phosphorylation of cMet and activation of cMet pathway through direct interaction. EGFL9 overexpression also increases ROS and lactate production, while decreasing COX activity and the oxygen consumption rate of the tumor cells expressing EGFL9. They conclude that EGFL9 induces a metabolic switch from oxidative phosphorylation to glycolysis. This paper uses strong and convincing data to support their conclusions. I have a few minor suggestions for the improvement of this paper before it gets accepted.

1. In figure 3B and 4D, the authors should include the number of metastatic nodules per lung beside the percentage the metastasis/lung.

Our response: We added the number of metastatic nodules per lung to the Figure 3b and 4d. Please see revised new Figures 3b, c and 4e, f

2. In figure 4C and D, the data will be better represented, if the authors could provide a bar graph instead of a table to show the difference in tumor size with error bars and significance values.

Our response: Thanks for this helpful suggestion. We have revised Figure 4c using bar graph to show the difference in tumor size. We also used dot plot to show the difference in metastatic numbers. Please see the revised Figure 4c and 4d.

3. In figure 5D, the changes in pAKT, pERK by western blot are not convincing and it is not adding any value to the conclusion. Either the authors could remove these two blots or repeat.

Our response: We repeated this Western blot experiment for pAKT and pERK, as well as their total protein counterparts. The results showed more apparent inhibition of pAKT and pERK,

while no apparent change in total proteins, through inhibition of cMET phosphorylation. Figure 5d was revised accordingly.

4. In figure 6B, the authors don't see a clear expression of endogenous cMet or EGFL9, which makes sense because 10% input may not be enough to see the endogenous proteins, but they do see a striking interaction using this pull down. However, one would expect for sure to see the overexpressed proteins in 10% input, but it is not the case. One possibility is that the panels are switched. The authors need to reconfirm these results by repeating it.

Our response: To address the reviewer's concern, we repeated these experiments by increasing the loading control from 10% to 20% of input in our revision. Under this condition, we are able to clearly demonstrate the expression of both endogenous cMET and EGFL9 in the lysates. Figures 6a and 6b were revised accordingly.

5. In figure 8K, the authors should be consistent in labeling single agent treatment in blue or the same color across.

Our response: According to the reviewer's suggestion, we changed the color presentation for the line of the treated cells to blue in the revised Figure 8k, which makes the color presentation consistent across the panels in the revised Figure 8.

Reviewer #3: Breast Cancer metastasis

(Remarks to the Author)

In the current study Meng et. al. identified that EGFL9 promotes breast cancer metastasis by inducing cMET activation and metabolic reprogramming. They also found that EGFL9 enhances stemness and chemoresistance in breast cancer. Moreover, targeting cMET signaling and metabolic reprogramming synergistically sensitizes breast cancer cells to chemotherapeutic agents. The findings described in this study are novel, interesting and clinically relevant. However, there are several minor points need to be addressed or discussed before publication.

1. In Fig.1, the mRNA levels of EGFL9 do not correlate well with protein levels across different cell lines. For example, MCF7 has higher mRNA than BT474 (Fig.1A), but lower protein expression (Fig.1B). It would be interesting to investigate the regulation (upstream) of EGFL9. Also, the authors should provide quantitative data for Fig.1B and C.

Our response: We appreciate the reviewer for pointing this out. The RT-PCR and Western blots have been repeated with fresh cells. We also provided quantitative data for both the RT-PCR and Western blot analysis. The results are presented in revised Figure 1b and c.

2. In Fig.1A, EGFL2 is highly expressed in luminal-like cells, which is opposite to EGFL9. It would be interesting to investigate or discuss to see if EGFL2 has opposite biological functions.

Our response: We agree it is very interesting that EGFL2¹⁷ exhibits a luminal expression pattern in contrast to the basal expression of EGFL9, which may suggest the differential roles of these two proteins in the regulation of breast cancer progression. While focusing on delineation of the novel role of EGFL9 in promoting breast cancer metastasis in current study, according to the reviewer's suggestion, we added discussion on the possible role of EGFL2 in breast cancer progression in light of its elevated expression in the luminal type. Due to the limited time available, we will not be able to experimentally explore this possibility. These discussions regarding EGFL2 are presented as a paragraph in the revised discussion section at page 16, line 425-430.

2. In Fig.S3A as well as the WB data in the following figures, it would be better to include EGFL9.

Our response: The EGFL9 expression was detected by V5 antibody. This result was added to the revised Supplementary Fig. S4a.

4. In Fig. 4C, why Sh3 has significant lower p-cMET and downstream signaling than Sh1? However, these two targets have similar EGFL9 knockdown efficacy (Fig. S2).

Our response: We have repeated the RT-PCR and the results were updated and matched to the cell signaling Western blotting results. See revised supplementary Figure S3g.

5. In Fig. 4D and E, given that p-AKT and p-ERK are not significantly inhibited by the cMET inhibitor JNJ, the treatment time points or concentrations of JNJ need to be further optimized. Also, EGFL9+JNJ treatment group has significant less migrated and invasive cells than LacZ+JNJ group, suggesting EGFL9 may have metastatic suppressive, rather than promoting roles in cells without cMET activation.

Our response: We believe that the previously insignificant inhibition of pAKT and pERK by the cMET inhibitor was due to the dysfunction of cMET inhibitor JNJ38877605. We thus repeated the treatment with freshly made JNJ38877605 in HMLE/EGFL9 cells and performed Western blot analyses. Our data showed that treatment with fresh JNJ38877605 inhibited pAKT and pERK1/2 in a dose dependent manner. Figure 5d was revised accordingly.

For the result shown in Figure 5e, our interpretation is that EGFL9 is a promoter of cell migration and invasion, because overexpression of EGFL9 alone was sufficient to significantly increase cell migration and invasion in comparison with HMLE/LacZ cells. In addition, treatment of HMLE/EGFL9 with JNJ38877605 showed more significant inhibition of migration and invasion than LacZ plus JNJ. HMLE/EGFL9 cells are more sensitive to JNJ than HMLE/LacZ cells because HMLE/EGFL9 cells have significantly increased cMET activation in comparison to HMLE/LacZ cells.

6. To confirm EGFL9 promotes metastasis specifically through cMET, downstream signaling should be examined in EGFL9 expressed/KD cells combined with JNJ treatment.

Our response: According to the reviewer's suggestion, to confirm the importance of cMET activation in mediating EGFL9 driving metastasis, we designed a new experiment. We first treated HMLE cells with cMET inhibitor JNJ and then cells were infected with lentivirus expressing EGFL9. By doing this, we tested if cMET activation is essential for EGFL9 to activate downstream signaling pathways related to metastasis. Our results indicated that infection of viruses containing EGFL9 significantly induce the phosphorylation of cMET and downstream EGFR and AKT. In contrast, in cells pre-treated with cMET inhibitor, EGFL9 expression could not significantly induce EGFR and AKT phosphorylation, suggesting cMET activation is critical for EGFL9 to drive metastasis-related signaling pathways. The new results were summarized in Supplementary Fig. S5c.

7. Total cMET cannot be observed in the input fractions in Fig. 6A and B. More input should be loaded.

Our response: We repeated all IP-Western blotting experiments by loading more than 20% input. Now cMET and EGFL9 can be observed in the input fractions in all the figure panels. Please see revised Figure 6a and b.

8. Previous study (Dai et. al., Blood 2015 126:2821-2831) suggested that cMET signaling is critical for cell cycle progression. Moreover, this study also suggested cMET is important for cell stemness. It would be better to discuss why EGFL9, which targets cMET, only inhibits metastasis but not primary tumor growth.

Our response: We thank the reviewer for raising this important question. Our understanding is that cMET function should be context-dependent and therefore may vary in different tissues, like many other genes or proteins. In terms of the function of EGFL9, although we only showed that it specifically activates the phosphorylation of cMET, EGFL9 may also modulate cell function through other unidentified mechanisms including protein-protein interactions. These functions

will eventually contribute to alterations in the cell context or tumor microenvironment, which may balance or modulate the effect of targeting cMET in EGFL9 overexpressing cells. We have discussed this issue in the discussion section of the revised manuscript in page 17-18, line 461-466 .

References for Reviewers.

1. Farmer P, *et al.* Identification of molecular apocrine breast tumours by microarray analysis. *Oncogene* **24**, 4660-4671 (2005).
2. Neve RM, *et al.* A collection of breast cancer cell lines for the study of functionally distinct cancer subtypes. *Cancer Cell* **10**, 515-527 (2006).
3. Tabchy A, *et al.* Evaluation of a 30-gene paclitaxel, fluorouracil, doxorubicin, and cyclophosphamide chemotherapy response predictor in a multicenter randomized trial in breast cancer. *Clin Cancer Res* **16**, 5351-5361 (2010).
4. Zhao H, *et al.* Different gene expression patterns in invasive lobular and ductal carcinomas of the breast. *Mol Biol Cell* **15**, 2523-2536 (2004).
5. Hidalgo M, *et al.* Patient-derived xenograft models: an emerging platform for translational cancer research. *Cancer Discov* **4**, 998-1013 (2014).
6. Ben-David U, *et al.* Patient-derived xenografts undergo mouse-specific tumor evolution. *Nat Genet* **49**, 1567-1575 (2017).
7. Paez-Ribes M, Man S, Xu P, Kerbel RS. Development of Patient Derived Xenograft Models of Overt Spontaneous Breast Cancer Metastasis: A Cautionary Note. *PLoS One* **11**, e0158034 (2016).
8. Willyard C. The mice with human tumours: Growing pains for a popular cancer model. *Nature* **560**, 156-157 (2018).
9. Kerppola TK. Bimolecular fluorescence complementation: visualization of molecular interactions in living cells. *Methods Cell Biol* **85**, 431-470 (2008).

10. Ohashi K, Kiuchi T, Shoji K, Sampei K, Mizuno K. Visualization of cofilin-actin and Ras-Raf interactions by bimolecular fluorescence complementation assays using a new pair of split Venus fragments. *Biotechniques* **52**, 45-50 (2012).
11. Pan JA, Ullman E, Dou Z, Zong WX. Inhibition of protein degradation induces apoptosis through a microtubule-associated protein 1 light chain 3-mediated activation of caspase-8 at intracellular membranes. *Mol Cell Biol* **31**, 3158-3170 (2011).
12. Clemente P, *et al.* hCOA3 stabilizes cytochrome c oxidase 1 (COX1) and promotes cytochrome c oxidase assembly in human mitochondria. *J Biol Chem* **288**, 8321-8331 (2013).
13. Dennerlein S, Rehling P. Human mitochondrial COX1 assembly into cytochrome c oxidase at a glance. *J Cell Sci* **128**, 833-837 (2015).
14. Ostergaard E, *et al.* Mutations in COA3 cause isolated complex IV deficiency associated with neuropathy, exercise intolerance, obesity, and short stature. *J Med Genet* **52**, 203-207 (2015).
15. Ding Y, *et al.* Receptor tyrosine kinase ErbB2 translocates into mitochondria and regulates cellular metabolism. *Nat Commun* **3**, 1271 (2012).
16. Kong-Beltran M, Stamos J, Wickramasinghe D. The Sema domain of Met is necessary for receptor dimerization and activation. *Cancer Cell* **6**, 75-84 (2004).
17. Vincent JB, Skaug J, Scherer SW. The human homologue of flamingo, EGFL2, encodes a brain-expressed large cadherin-like protein with epidermal growth factor-like domains, and maps to chromosome 1p13.3-p21.1. *DNA Res* **7**, 233-235 (2000).

Reviewers' comments:

Reviewer #2 (Remarks to the Author):

Authors have addressed all my comments/suggestions.

Reviewer #3 (Remarks to the Author):

The authors have addressed the points raised during the last round of review with new data and clarification, and improved the quality of the manuscript. It is acceptable for publication now.

Reviewer #4 (Remarks to the Author):

This study investigates the role of EGFL9 in breast cancer cells metastasis. It is shown that EGFL9 fosters metastasis in vitro and in vivo with no change in cell proliferation, but the mechanism does not involve changes in EMT markers. Rather, stimulation of stemness is found. Then, activation of C-met is observed and a physical interaction between EGFL9 and C-met is discovered and the complex is found in mitochondria. While some parts of this study are convincing (kinase screening; co-IPs) the cancer biology part remains unclear as regard to the mechanisms underpinning metastasis and the relationships with bioenergetics.

Major issues:

1-it is not clear how EGFL9-Cmet fosters metastasis and how the observed increased metastasis in the lung relates to the improvement of stemness in absence of EMT stimulation.

2- how do EGFL9 and C-met enter the mitochondrion and then co-associate is not explained. The EGFL9 IP-mitochondrial partners are 'data not shown'; why ?

I was convinced that EGFL9-cMET signaling can phosphorylate COX and impair OXPHOS but the localization of EGFL9-CMET complex within the mitochondrion is not clearly demonstrated. EM studies are required as well as experiments in cells lacking functional mitochondria as TFAM KO. Does EGFL9 overexpression maintain or increase its metastatic potential in OXPHOS deficient cells ? Also, several studies showed that metastasis associates more with OXPHOS while increased proliferation is linked with the Warburg effect. Here, the authors find the opposite and such contradiction is interesting but requires firm demonstration. Also, the use of 2DG as inhibitor of glycolysis is not correct because 2DG also blocks pyruvate-supported OXPHOS, weakening the bioenergetic conclusions. The cell culture in 25mM glucose media can also bias the conclusions and drives the bioenergetic phenotype toward glycolysis. Tumors have 5mM glucose or even lower levels.

COX activity is measured indirectly using oxygraphy. Directly enzymatic measurement should be made in the context of this study to avoid membrane effect, respirasome effect and endogenous cytc c effect.

mt-membrane potential analysis: methods section, the probe used is not given.

ATP assay: 2 hours treatment with rotenone is too long and cells with compensate for OXPHOS inhibition using the Pasteur effect. short time treatment are required to evaluate mt-ATP.

Minor:

#1: A study on the TCGA cohort is required to evaluate the impact of EGFL9 expression on patients survival and metastatic stage to extend the observations on a larger cohort.

#2: groups of 5 animals are used and this is poor to derive strong statistics.

Responses to reviewers' comments:

Reviewer #2:

Authors have addressed all my comments/suggestions.

Response: We thank the reviewer for his/her time in reviewing our manuscript and the invaluable comments and suggestions.

Reviewer #3:

The authors have addressed the points raised during the last round of review with new data and clarification, and improved the quality of the manuscript. It is acceptable for publication now.

Response: We sincerely thank the reviewer for the positive comments on our manuscript and the recommendation for publication.

Reviewer #4:

This study investigates the role of EGFL9 in breast cancer cells metastasis. It is shown that EGFL9 fosters metastasis in vitro and in vivo with no change in cell proliferation, but the mechanism does not involve changes in EMT markers. Rather, stimulation of stemness is found. Then, activation of C-met is observed and a physical interaction between EGFL9 and C-met is discovered and the complex is found in mitochondria. while some parts of this study are convincing (kinase screening; co-IPs) the cancer biology part remains unclear as regard to the mechanisms underpinning metastasis and the relationships with bioenergetics.

Response: We are deeply appreciative of the reviewer's comments and suggestions to improve the scientific merit of the manuscript. Below please find our point-by-point response to these comments.

Major issues:

1. It is not clear how EGFL9-Cmet fosters metastasis and how the observed increased metastasis in the lung relates to the improvement of stemness in absence of EMT stimulation.

Response: Our study demonstrates that EGFL9 is a novel metastasis-driving gene. Towards understanding EGFL9 function in driving cancer metastasis, we examined two partially independent mechanisms of EGFL9: 1) cMET activation; and 2) metabolic reprogramming.

cMET activation has been observed to promote metastasis in several cancer types, including breast, and inhibition of cMET was observed to cause a decrease in cancer metastasis to visceral organs and the bone^{1, 2}. These studies also observed an increase in cancer stemness, consistent with our findings. However, these studies focused on canonical cMET ligand, HGF. Here, we report EGFL9-mediated cMET activation as a novel mechanism to confer cancer stem-like traits. By using the cMET inhibitor JNJ, we confirmed that cMET activation is required for EGFL9 driven metastasis-associated *in vitro* characteristics of cell migration/invasion and stemness. These results strongly suggest that EGFL9-mediated cMET activation is at least partially responsible for the pro-metastatic effect of EGFL9.

In parallel, accumulated evidence in recent years has suggested that metabolic reprogramming and the Warburg phenotype promotes metastasis^{3, 4, 5}. In addition, mitochondrial dysfunction is also closely correlated with tumor progression and metastasis^{6, 7}. In this report, we observed EGFL9 led to mitochondrial dysfunction (interaction with COA3 and decrease in COX activity) and metabolic reprogramming (decrease in OXPHOS, and increase in lactate production and glucose uptake). While this study did not specifically examine the role of metabolic reprogramming in EGFL9-mediated metastasis, we demonstrate that treatment with glycolysis inhibitors decreases EGFL9-induced stemness, suggesting EGFL9-induced dysregulation of bioenergetics is important for EGFL9 promoted metastasis.

We fully understand the importance of studying how the interaction of EGFL9-cMET contributes to cancer metastasis. However, precisely deciphering the structure of this interaction and identifying specific small molecular compounds to target this interaction will take years of work. We are currently applying for a grant to explore this direction. We have implemented this discussion in our manuscript (lines 502-505, page 20).

Although EMT has been suggested as a driving force to cancer metastasis, several recent studies have demonstrated that EMT is not necessarily required for metastasis^{8, 9}. In addition, there are two major cancer stem-like cell sub-populations in human breast cancer: ALDH⁺ and CD44⁺/CD24⁻. The results

from our research group and others show that these two subpopulations are not identical^{10, 11}. The ALDH⁺ stem cell subpopulation is proliferative and remains in an epithelial-like state, while the CD44⁺/CD24⁻ subpopulation is quiescent, invasive, and mesenchymal-like. This is consistent with other studies showing the EMT is closely related with CD44⁺/CD24⁻. In this study we observe that EGFL9 expression results in enrichment of ALDH⁺, not the CD44⁺/CD24⁻, stem-like cell subpopulation, suggesting that EGFL9 enhances stemness and metastasis through an EMT-independent manner. We have added this clarification to the discussion section of our manuscript (lines 453-460, page 18).

2. How do EGFL9 and C-met enter the mitochondrion and then co-associate is not explained. The EGFL9 IP-mitochondrial partners are data not show; why?

Response: In this manuscript we are focusing on: (1) identifying and functionally characterizing the novel metastasis-driving gene EGFL9 in breast cancer, (2) the ability of EGFL9 to activate cMET signaling, (3) the EGFL9-mediated regulation of metabolism, and (4) the effect of EGFL9 in driving cancer stemness and potential therapeutic strategy. While revealing how EGFL9 and cMET translocate into mitochondria is relevant and important, we believe this mechanistic exploration is beyond the scope of current study and will serve as a future direction.

The initial reason for us to perform EGFL9-IP in mitochondria was to identify an EGFL9 mitochondrial binding partner to use for the BiFC assay. Based on that result, we identified COA3 as a potential EGFL9 interacting partner and performed BiFC to confirm EGFL9 and COA3 interaction. COA3 is a well-known protein localizing within the mitochondrial inner membrane. The results provide strong evidence to support the localization of EGFL9 in the mitochondria, given that the BiFC signal can only be reconstituted when the two proteins are within 100 angstroms (10nm)^{12, 13}. Since the other mitochondrial partners will be further explored in future studies, and as they are not critical to the aims of this study, we decided not to release the whole list of EGFL9 co-IP proteins at this time.

3. I was convinced that EGFL9-cMET signaling can phosphorylate COX and impair OXPHOS but the localization of EGFL9-CMET complex within the mitochondrion is not clearly demonstrated. EM

studies are required as well as experiments in cells lacking functional mitochondria as TFAM KO. Does EGFL9 overexpression maintains or increase its metastatic potential in OXPHOS deficient cells?

Response: In our first submission, we confirmed EGFL9-cMET interaction with immunofluorescence-confocal microscopy assay and Co-IP/Western blotting. Also, I believe we cited a paper previously ¹⁴, which independently determined that cMET can localize to the mitochondria. However, the original reviewer 1 was concerned that our methods were based entirely on antibodies. In our revision, we performed BiFC and proteinase protection analysis to further confirm EGFL9 /cMET complex localization in mitochondria. As mentioned above, the BiFC assay is an antibody-free method and the signal can only be generated when the two interaction partners are within 100 angstroms ^{12, 13}. Based on these results, we believe the localization of this complex in mitochondria has been clearly demonstrated.

We agree EM could provide a higher resolution technique to further support the localization of these proteins in mitochondria. However, this technique is still limited by antibody specificity. Unfortunately, the Electronic Microscopy core facility is not available for us at either our home institution (Wayne State University) or other nearby facilities (Michigan State University, University of Michigan). We have contacted several core facilities around USA; however, most of them requires several months waiting time for doing this kind of work.

[Redacted]

[Redacted]

4. Also, several studies showed that metastasis associates more with OXPHOS while increased proliferation is linked with the Warburg effect. Here, the authors find the opposite and such contradiction is interesting but requires firm demonstration.

Response: We agree there is controversy regarding the association between metabolic status and metastasis. The Warburg metabolic phenotype has indeed been linked to increased cell proliferation in certain contexts. However, accumulated evidence in recent years has suggested that metabolic reprogramming and Warburg effect promotes metastasis^{3, 4, 5}. In addition, mitochondrial dysfunction was also demonstrated to be closely correlated with tumor progression and metastasis^{6, 7}. Our study provides evidence indicating that EGFL9 can promote metabolic reprogramming (as evidenced by decrease in OXPHOS, and increase in lactate production and glucose uptake), along with cMET activation. Additionally, we demonstrated that EGFL9 expression led to an enrichment of the ALDH⁺ stem cell population, which is both highly stem-like and highly proliferative. Moreover, treatment of cells with EGFL9 overexpression using glycolysis inhibitor (2-DG or NHI-2) results in decrease of ALDH⁺ subpopulation and mammosphere formation, suggesting metabolic reprogramming is essential for maintaining the stemness of EGFL9 driving metastatic cells. Therefore, our data is consistent with these recent discoveries, suggesting metabolic reprogramming or a metabolic plasticity contributes to cancer metastatic phenotype in our research context. We have implemented this portion of discussion in our manuscript in lanes 463-474, page 18-19.

5. Also, the use of 2DG as inhibitor of glycolysis is not correct because 2DG also blocks pyruvate-supported OXPHOS, weakening the bioenergetic conclusions. The cell culture in 25 mM glucose media can also bias the conclusions and drives the bioenergetic phenotype toward glycolysis. Tumors have 5mM glucose or even lower levels.

Response: We fully agree with the reviewer's comment that 2-DG is not a specific glycolysis inhibitor. However, most drugs currently used are not specific. More importantly, the cell model used exhibits increased glycolysis and decreased OXPHOS. We treated this cell model with a glycolysis inhibitor in order to observe the effect of glycolysis inhibition on cell stemness. To address the concern of about specificity of 2-DG, we choose another glycolysis inhibitor, NHI-2, and repeated these experiments. NHI-2 is an inhibitor of Lactate dehydrogenase-A (LDH-A, LDHA), a key enzyme necessary to sustain glycolysis, the major pathway used by many cancer cells for cell growth and proliferation (the Warburg effect). NHI-2 has anti-glycolytic activity in a variety of cancer cells. In contrast, 2-DG (2-Deoxyglucose) is a glucose analog that inhibits glycolysis via its actions on hexokinase.

Our results demonstrated that treatment of cells with NHI-2 showed a dose-dependent inhibitory effect on ALDH⁺ subpopulation. This effect reproduces the results we observed for 2-DG treatment. Consistent with this result, treatment of NHI-2 also significantly impaired the tumorsphere formation capability of HMLE/EGFL9 cells. This result, together with 2-DG, suggest that targeting glycolysis is an effective approach to impair the stemness of cancer cells, which may contribute to the metastasis inhibition for cancer treatment. We have added these results as new supplementary Figure 12.

We thank for the reviewer for raising the question regarding glucose concentration in cell culture. We realized our previous description may have been confusing. Actually, the high glucose (4500 mg/L) DMEM medium is used in 4T1 cell culture. For studying the bioenergetics switch, we used low glucose DMEM medium (1000 mg/L) will be mixed with F12 in 1:1 ratio for the HMLE cell model. The final concentration of glucose in this medium is only 500 mg/L. We have revised the manuscript and make this issue clear in the material and methods section of cell culture (lines 520-525, page 20-21).

6. COX activity is measured indirectly using oxygraphy. Directly enzymatic measurement should be made in the context of this study to avoid membrane effect, respirasome effect and endogenous cytc c effect.

Response: To address this concern, we performed the experiment to directly measure the COX activity in HMLE/LACZ and /EGFL9 cell lines using a Cytochrome Oxidase Activity Colorimetric Assay Kit” from Biovision. We observed approximately two-fold higher cytochrome oxidation in mitochondria isolated from the HMLE/LACZ cell line, compared with HMLE/EGFL9. This difference was statistically significant ($P= 0.03$). This data is consistent with the effect we obtained using the oxygen electrode system. We replace the previous result with the new COX activity result (Figure 7c). We also update the method with COX Activity Assay on page 29.

7. mt-membrane potential analysis: methods section, the probe used is not given.

Response: We have already added JC-1 as the probe in the method section. Please check line 784, on page 31.

8. ATP assay: 2 hours treatment with rotenone is too long and cells with compensate for OXPHOS inhibition using the Pasteur effect. short time treatments are required to evaluate mt-ATP.

Response: We thank the reviewer for raising the insightful concern. We changed the treatment time to 30mins and repeat the experiments. Our data showed that cells treated with Oligomycin (10 μ M) or Rotenone (10 μ M) for 30 mins showed a more moderate effect, but reproduced the same trend as the effect observed for cells treated for 2hrs. ATP production maintained at high level after OXPHOS inhibition in HMLE/EGFL9 cells in comparison with control HMLE/LacZ cells, suggesting a great portion of ATP production is attributed to glycolysis in cells with ectopic expression of EGFL9. The new figure replaced the old one in Supplementary Figure 11d.

Minor questions:

#1: A study on the TCGA cohort is required to evaluate the impact of EGFL9 expression on patient survival and metastatic stage to extend the observations on a larger cohort.

#2: groups of 5 animals are used and this is poor to derive strong statistics.

Response: As suggested by editor, we skipped the two Minor questions.

References for Reviewers

1. D'Amico L, *et al.* C-met inhibition blocks bone metastasis development induced by renal cancer stem cells. *Oncotarget* **7**, 45525-45537 (2016).
2. Xing F, *et al.* Activation of the c-Met Pathway Mobilizes an Inflammatory Network in the Brain Microenvironment to Promote Brain Metastasis of Breast Cancer. *Cancer Res* **76**, 4970-4980 (2016).
3. Lehuède C, Dupuy F, Rabinovitch R, Jones RG, Siegel PM. Metabolic Plasticity as a Determinant of Tumor Growth and Metastasis. *Cancer Res* **76**, 5201-5208 (2016).

4. Lu J. The Warburg metabolism fuels tumor metastasis. *Cancer Metastasis Rev* **38**, 157-164 (2019).
5. Porporato PE, *et al.* A mitochondrial switch promotes tumor metastasis. *Cell Rep* **8**, 754-766 (2014).
6. Chen EI. Mitochondrial dysfunction and cancer metastasis. *J Bioenerg Biomembr* **44**, 619-622 (2012).
7. Hsu CC, Tseng LM, Lee HC. Role of mitochondrial dysfunction in cancer progression. *Exp Biol Med (Maywood)* **241**, 1281-1295 (2016).
8. Fischer KR, *et al.* Epithelial-to-mesenchymal transition is not required for lung metastasis but contributes to chemoresistance. *Nature* **527**, 472-476 (2015).
9. Zheng X, *et al.* Epithelial-to-mesenchymal transition is dispensable for metastasis but induces chemoresistance in pancreatic cancer. *Nature* **527**, 525-530 (2015).
10. Liu S, *et al.* Breast cancer stem cells transition between epithelial and mesenchymal states reflective of their normal counterparts. *Stem Cell Reports* **2**, 78-91 (2014).
11. Wu L, *et al.* Disulfiram and BKM120 in Combination with Chemotherapy Impede Tumor Progression and Delay Tumor Recurrence in Tumor Initiating Cell-Rich TNBC. *Sci Rep* **9**, 236 (2019).
12. Harmon M, Larkman P, Hardingham G, Jackson M, Skehel P. A Bi-fluorescence complementation system to detect associations between the Endoplasmic reticulum and mitochondria. *Sci Rep* **7**, 17467 (2017).
13. Kerppola TK. Bimolecular fluorescence complementation (BiFC) analysis as a probe of protein interactions in living cells. *Annu Rev Biophys* **37**, 465-487 (2008).
14. Guo T, *et al.* Quantitative proteomics discloses MET expression in mitochondria as a direct target of MET kinase inhibitor in cancer cells. *Mol Cell Proteomics* **9**, 2629-2641 (2010).
15. Lu W, *et al.* Novel role of NOX in supporting aerobic glycolysis in cancer cells with mitochondrial dysfunction and as a potential target for cancer therapy. *PLoS Biol* **10**, e1001326 (2012).

16. Pelicano H, *et al.* Mitochondrial dysfunction in some triple-negative breast cancer cell lines: role of mTOR pathway and therapeutic potential. *Breast Cancer Res* **16**, 434 (2014).

REVIEWERS' COMMENTS:

Reviewer #4 (Remarks to the Author):

The authors could not address my comments.

How the interaction between EGFL9-cMET and bioenergetics contributes to metastasis remain unexplained. More important, how EGFL9 and cMET translocate to mitochondria is central to this study and also remains unexplained. Lastly, the point on 5 animals per group is not minor as statistics are essential to address the relevance of translational studies in Nature Communication. The editor of Nature Communication cannot neglect this point.

Validation of the findings on another mouse model is also typically requested by the journal to strengthen the conclusions and this is missing here.

So, the findings have not been sufficiently strengthened at a mechanistic level to address my previous concerns

Responses to reviewers' comments:

Reviewer #4:

The authors could not address my comments.

a) How the interaction between EGFL9-cMET and bioenergetics contributes to metastasis remain unexplained. b) More important, how EGFL9 and cMET translocate to mitochondria is central to this study and also remains unexplained. c) Lastly, the point on 5 animals per group is not minor as statistics are essential to address the relevance of translational studies in Nature Communication. The editor of Nature Communication cannot neglect this point. d) Validation of the findings on another mouse model is also typically requested by the journal to strengthen the conclusions and this is missing here. So, the findings have not been sufficiently strengthened at a mechanistic level to address my previous concerns

We are deeply appreciative of the reviewer's comments to improve the scientific merit of the manuscript. Below please find our point-by-point response to these comments.

Response to comment a) How the interaction between EGFL9-cMET and bioenergetics contributes to metastasis remain unexplained.

We have shown that EGFL9 can drive cell migration/invasion *in vitro* and cancer metastasis *in vivo*. Two potential mechanisms underlying the EGFL9 metastasis-driving function were revealed. First, EGFL9 expression leads to activation of the cMET signaling pathway, which is well-acknowledged to promote cancer cell motility. 2) EGFL9 expression also leads to a metabolic reprogramming, which results in enhancement of cancer stemness, another factor contributing to cancer metastasis.

Interestingly, we also observed that EGFL9 interacts with cMET in TNBC cells. However, how the interaction of EGFL9-cMET contributes to cancer metastasis remained unclear. We speculate that the EGFL9-cMET interaction facilitates cMET activation as a major factor to promote cancer metastasis. This hypothesis is based on two observations. First, we showed specific activation of cMET upon EGFL9 expression and cMET activation has been reported to contribute to cancer metastasis^{1,2}. Second, cell surface molecular interactions have been frequently reported as a mechanism to activate RTKs, including EGFR and cMET^{3,4}. In future studies, we will focus on precisely deciphering the structure of EGFL9-cMET interaction and identifying critical fragments or amino acids in EGFL9 protein using a

combined genetic and biochemical approach. The mutant EGFL9, which is incapable of interacting with cMET, will then be examined for its effect on cMET activation and distant metastasis. This part of discussion is added to the text from lines 464 to 474, pages 18-19.

Cancer stemness has been well recognized as a contributor to cancer metastasis (refs). Our data showed that EGFL9 expression leads to enrichment of ALDH⁺ stem cell subpopulation and increase of tumorsphere formation. This effect can be reversed by glycolysis inhibition. Therefore we believe that EGFL9-induced bioenergetics contributes to cancer metastasis through enhancing cancer stemness. We have added this clarification in the discussion section of our manuscript (lines 440-447, pages 17-18).

Response to comment b) More important, how EGFL9 and cMET translocate to mitochondria is central to this study and also remains unexplained.

Accumulating evidences show that endocytosis is the major regulator of signaling from receptor tyrosine kinases (RTKs) ⁵. The canonical model of RTK endocytosis involves rapid internalization of an RTK activated by ligand binding at the cell surface and subsequent sorting of internalized ligand-RTK complexes to lysosomes for degradation. In addition, translocation of these RTKs to the nucleus and mitochondria has also been reported. Specifically, EGFR has been shown to translocate to mitochondria through both endocytosis dependent and independent mechanisms ^{6, 7}. Her2 protein was reported to translocate into mitochondria by interacting with HSP70 ⁸. In the future, we will explore if EGFL9 and cMET translocate to mitochondria via an endocytosis-linked mechanism and whether this translocation is chaperone dependent. We have added this part of discussion in our revised manuscript (lines 476-486, page 19).

Response to Comment c) The point on 5 animals per group is not minor as statistics are essential to address the relevance of translational studies in Nature Communication. The editor of Nature Communication cannot neglect this point.

We collaborated with Dr. Wei Chen, a statistician from the Karmanos Cancer Institute Biostatistics Core, to ensure appropriate experimental design, endpoints definitions, analysis plan, interpretation, and reporting of results since 2013. Sample size of biological replicates for *in vivo* study was estimated based on the hypothesis testing of primary endpoint using statistical sample size software PASS®.

One of the major primary endpoints in this study is the percentage of metastatic tumor area at the time of euthanasia. For each mice model, comparisons between the two conditions were done with two-sided two-sample t test. Wilcoxon rank sum test will be used if normality assumption doesn't hold after necessary transformation for normality.

Based on our published data ⁹, it was estimated that the mean and ranges (min, max) in the control and knockdown groups using 4T1/Balb/c model are about 40% (20%, 60%) and 10% (5%, 25%), respectively. The estimated standard deviation (SD) in the control and knockdown groups with sample size of 5 each are 0.177 and 0.097, respectively. Group sample size of 5 achieves 80% power at significance level of 0.05 to detect a difference of 30% with unequal SD using two-sided two-sample t test.

This is a large effect size, which is justified by our previous experience. Due to ethical considerations, we chose to limit the number of mice to achieve the statistical significance. The results of current studies demonstrated that both EGFL9 overexpression and knockdown significantly affect the metastasis of cancer cells, which is consistent with our power analysis. Thus, it is statistically unnecessary to use more mice to demonstrate the effect of EGFL9 on metastasis in our study.

Response to comment d) Validation of the findings on another mouse model is also typically requested by the journal to strengthen the conclusions and this is missing here.

Thanks for the reviewer to bring up this question. Actually, in this manuscript, we used two different mouse models to demonstrate the effect of EGFL9 on cancer metastasis *in vivo*. We first implanted EpRas cells with ectopic expression of EGFL9 in the fat pads of NCR nu/nu mice and observed increased metastasis upon EGFL9 expression (Figure 3). This experiment demonstrates that EGFL9 is

sufficient to increase metastatic capability. To confirm this result, we implanted 4T1 cells with EGFL9 knockdown in the fat pads of BALB/c mice and observed decreased metastasis pattern upon EGFL9 knockdown (Figure 4). This second approach validates that EGFL9 is necessary for maintaining metastatic potential. We also used a third cell model, SUM 159, to validate our findings *in vitro*. However, as SUM159 is not capable of forming spontaneous metastasis *in vivo*, we did not use this model for metastasis studies.

References

1. Gherardi E, Birchmeier W, Birchmeier C, Vande Woude G. Targeting MET in cancer: rationale and progress. *Nat Rev Cancer* **12**, 89-103 (2012).
2. Spina A, *et al.* HGF/c-MET Axis in Tumor Microenvironment and Metastasis Formation. *Biomedicines* **3**, 71-88 (2015).
3. Velpula KK, Dasari VR, Asuthkar S, Gorantla B, Tsung AJ. EGFR and c-Met Cross Talk in Glioblastoma and Its Regulation by Human Cord Blood Stem Cells. *Transl Oncol* **5**, 379-392 (2012).
4. Viticchie G, Muller PAJ. c-Met and Other Cell Surface Molecules: Interaction, Activation and Functional Consequences. *Biomedicines* **3**, 46-70 (2015).
5. Goh LK, Sorkin A. Endocytosis of receptor tyrosine kinases. *Cold Spring Harb Perspect Biol* **5**, a017459 (2013).
6. Demory ML, *et al.* Epidermal growth factor receptor translocation to the mitochondria: regulation and effect. *J Biol Chem* **284**, 36592-36604 (2009).
7. Yao Y, *et al.* Mitochondrially localized EGFR is independent of its endocytosis and associates with cell viability. *Acta Biochim Biophys Sin (Shanghai)* **42**, 763-770 (2010).
8. Ding Y, *et al.* Receptor tyrosine kinase ErbB2 translocates into mitochondria and regulates cellular metabolism. *Nat Commun* **3**, 1271 (2012).
9. Zhang H, *et al.* Engagement of I-branching β -1, 6-N-acetylglucosaminyltransferase 2 in breast cancer metastasis and TGF- β signaling. *Cancer Res* **71**, 4846-4856 (2011).